# Experimental evaluation of the importance of colonization history in early-life gut microbiota assembly

Inés Martínez[1,2], Maria X Maldonado-Gomez[1], João Carlos Gomes-Neto[1], Hatem Kittana[1], Hua Ding[3], Robert Schmaltz[1], Payal Joglekar[3], Roberto Jiménez Cardona[1], Nathan L Marsteller[1], Steven W Kembel[4], Andrew K Benson[1], Daniel A Peterson[1,3,5], Amanda E Ramer-Tait[1], Jens Walter[1,2,6]*

[1]Department of Food Science and Technology, University of Nebraska-Lincoln, Lincoln, United States; [2]Department of Agricultural, Food and Nutritional Science, University of Alberta, Edmonton, Canada; [3]Department of Pathology, Johns Hopkins University School of Medicine, Baltimore, United States; [4]Département des sciences biologiques, Université du Québec à Montréal, Montreal, Canada; [5]Eli Lilly & Co, Indianapolis, United States; [6]Department of Biological Sciences, University of Alberta, Edmonton, Canada

**Abstract** The factors that govern assembly of the gut microbiota are insufficiently understood. Here, we test the hypothesis that inter-individual microbiota variation can arise solely from differences in the order and timing by which the gut is colonized early in life. Experiments in which mice were inoculated in sequence either with two complex seed communities or a cocktail of four bacterial strains and a seed community revealed that colonization order influenced both the outcome of community assembly and the ecological success of individual colonizers. Historical contingency and priority effects also occurred in $Rag1^{-/-}$ mice, suggesting that the adaptive immune system is not a major contributor to these processes. In conclusion, this study established a measurable effect of colonization history on gut microbiota assembly in a model in which host and environmental factors were strictly controlled, illuminating a potential cause for the high levels of unexplained individuality in host-associated microbial communities.
DOI: https://doi.org/10.7554/eLife.36521.001

*For correspondence:
jwalter1@ualberta.ca

## Introduction

The human gastrointestinal microbiota makes essential contributions to host metabolic, physiological, and immune functions (*Fujimura et al., 2010*), and aberrations in its structure contribute to pathologies and chronic disease states (*Houghteling and Walker, 2015*). Microbiome assembly begins at birth and undergoes dynamic processes influenced by factors such as birth method, feeding method, and antibiotic treatment (*Bokulich et al., 2016*). Once established, the community is dominated by a core set of species characteristic to the host species and is resilient to perturbations (*Dethlefsen and Relman, 2011*; *Martínez et al., 2013*; *Schloissnig et al., 2013*; *David et al., 2014a*). Despite the profound importance of the host-microbiome symbiosis for human health, the gut microbiota is characterized by a high degree of variation in composition and relative abundance of community members, which is referred to as ß-diversity (*Huttenhower, 2012*). This individuality can relate to functions important for health (e.g. the bio-conversion and activation of dietary compounds and drugs), and it is likely that misconfigurations play a role in disease predisposition (*Vatanen et al., 2016*; *Kostic et al., 2015*; *Koh et al., 2016*; *Spanogiannopoulos et al., 2016*).

**eLife digest** The microbial community living in the gastrointestinal tract of humans, also known as the gut microbiome, is essential for health. Disturbances of this community can lead to chronic diseases. Each person has a unique and stable community of gut microbes that is as personal as a 'fingerprint'. Studies have shown that an individual's genetics, diet, environment, lifestyle, and physiological state all make small contributions to the variation of the gut microbiome among individuals. However, less than 30% of this variation can be explained, and even identical twins, who share the same genetics and often diets and lifestyle, have distinct gut microbiomes. This suggests that other unknown factors likely shape these microbial communities too.

The microbial communities and the gut make up an ecosystem that is likely subject to many of the same ecological rules that govern ecosystems like rainforests or coral reefs. Yet many studies have overlooked the role of ecology in shaping the gut microbiota. For example, it is well known that the order in which organisms arrive in a community may influence how they interact and assemble into communities. It is possible that the order bacteria are introduced into the gastrointestinal tract of babies early in life may also change the make up of their gut microbiome, and thus introduce the variation that is currently unaccounted for.

Now, Martínez et al. show that the first types of bacteria to colonize the gut of mice have a lasting impact on their microbiome. In the experiments, genetically identical mice were housed under exactly the same conditions in airtight plastic bubbles. This allowed the scientists to control when the young mice first encountered specific microbes and microbe communities. Distinct microbial communities collected from different adult mice were introduced into the gastrointestinal tract of the young mice in sequence. Martínez et al. found that the microbes they introduced into the young mice first had the strongest influence on their gut microbiome at the end of the experiments.

When the experiments were repeated with a cocktail of four different bacteria the results were similar – the earlier arrivals showed enhanced colonization and had the biggest influence on the microbe community. This suggests that the timing of bacterial arrival in the gut is very important to shape the gut microbiome. Since it is highly random and unpredictable in real-life, and likely to differ even among twins, it could explain why the gut microbiome can be so unique. More studies are needed to understand how antibiotics, formula feeding, or cesarean sections affect gut microbiota early in life, and consequently health. This may help scientists develop better ways to influence the microbiota to improve health, for example, by introducing beneficial microbes early in life.

DOI: https://doi.org/10.7554/eLife.36521.002

Given these associations, a critical need exists to define the fundamental ecological principles that govern and regulate gut microbiome structure and the mechanisms that drive microbiome variation across hosts.

To date, research aimed at identifying the factors that cause microbiome variation has focused largely on host genetics and diet. Host genetic variation has an established role in driving β-diversity in both mice (*Benson et al., 2010*) and humans (*Turpin et al., 2016*; *Wang et al., 2016*) and a sub-set of fecal bacterial taxa has been shown to be partly heritable and/or associated with host single-nucleotide polymorphisms or quantitative trait loci (*Turpin et al., 2016*; *Wang et al., 2016*; *Goodrich et al., 2016*; *Benson, 2016*). However, most taxa do not show any significant heritability, and for those that do, heritability estimates are generally low compared to other heritable traits (*Goodrich et al., 2016*; *Hall et al., 2017*; *Goodrich et al., 2017*). In addition, genetic studies in humans show a low degree of repeatability, with only a small number of microbiome genome-wide associations being reproducible (*Turpin et al., 2016*; *Hall et al., 2017*; *Goodrich et al., 2017*). Even the combined effects of host genotype and dietary variables often do not account for the overall variation in a given taxonomic 'trait' (*Benson, 2016*; *Leamy et al., 2014*). The rather low contribution of host genetics to microbiome variation and the lack of heritability across many taxa is reflected in studies on monozygotic twins, which although slightly more similar than dizygotic twins, exhibit a substantial degree of individuality (*Goodrich et al., 2016*; *Goodrich et al., 2014*). In mice, the effect

of diet on microbiome structure has been shown to exceed the contribution of host genetics (*Carmody et al., 2015*). However, the contribution of diet to human microbiome variation was estimated to account for only around 6% (*Wang et al., 2016*), and standardizing the diet did not reduce β-diversity across study participants (*Wu et al., 2011*; *David et al., 2014b*). Although additional factors such as lifestyle (e.g. smoking), host physiology (e.g. age), and medication contribute to microbiome variation, non-genetic and genetic factors each account for approximately 10% of the variation in gut microbiota and thus fail to explain the majority of the inter-individual diversity that is observed in human cohorts (*Wang et al., 2016*; *Falony et al., 2016*; *Rothschild et al., 2018*).

When attempting to understand compositional features of the gut microbiota, one must consider that the studies performed to date have not accounted for the full range of ecological processes predicted to shape community diversity (*Walter, 2015*). Factors such as genotype, diet, smoking, medication, and host physiology all contribute to a deterministic process (or 'selection') that impacts diversity through niche-based mechanisms, thus selecting for specific taxa based on fitness differences (*Vellend, 2010*). However, according to ecological theory, variation among local communities is not only shaped by selection but also by historical and neutral processes (*Cavender-Bares et al., 2009*), which can only be described in probabilistic terms and include stochastic events such as ecological drift that occur randomly with respect to species fitness differences (*Hubbell, 2001*). With respect to historical processes, ecological theory holds that the timing and order of species immigration during community assembly can cause variation in the structure and function of communities, or historical contingency, through priority effects and monopolization (*Fukami, 2015*; *De Meester et al., 2016*). These effects could be paramount in shaping microbiomes as community assembly is initiated for each newborn since mammals are born germ-free (*Perez-Muñoz et al., 2017*). After birth, the offspring gradually acquires a microbial population, and community assembly occurs in parallel to host immunological, physiological, and metabolic maturation, with reciprocal interactions between microbes and host postnatal development (*Arrieta et al., 2014*). It is increasingly recognized that an early-life 'window of opportunity' is a critical period for both microbiome assembly and microbiome-driven host development (*Hansen et al., 2013*; *Lafores-Lapointe and Arrieta, 2017*). Priority effects have been suggested to influence gut microbiota assembly (*Sprockett et al., 2018*), and circumstantial evidence as well as the characteristics of the adult gut microbiota (stability and resilience despite pronounced inter-individual variation) are consistent with a combined effect of niche-related (deterministic) and early-life historical processes on community assembly (*Walter and Ley, 2011*). However, experimental evidence for the importance of colonization history in the gut microbiota assembly is lacking.

In this study, we employed a carefully controlled experimental model that allowed us to elucidate the importance of colonization history in early-life gut microbiota assembly by varying arrival order and timing of complex seed microbial communities in previously germ-free mice. The effect of arrival timing on the colonization success of specific gut bacteria and its impact on assembly trajectory of the resident community was also assessed. In both approaches, the influence of the host's adaptive immune system in causing priority effects and historical contingency was determined.

## Results

### Establishing a mouse model using distinct, complex microbial communities to study the importance of colonization history

To determine whether colonization history affects early-life gut microbiota assembly, experiments were first conducted to test the importance of arrival order when colonizing germ-free C57BL/6 mice with two compositionally distinct complex cecal microbial communities obtained from adult mice (*Mus musculus domesticus*). Experiments consisted of three different treatment groups. In two of the treatments, donor communities were sequentially inoculated in alternating order (*Figure 1*), with one treatment group being inoculated first with community A and second with B (group A/B) and the second treatment group being first inoculated with community B followed by community A (group B/A). In a third treatment group, both donor communities were inoculated at each time point (group AB/AB). The age of 10 days was chosen for the first inoculation as it was the earliest age at which pups could be gavaged without jeopardizing their health or parental care. At weaning (21 days), parent mice were sacrificed and their ceca collected (samples Parents A, Parents B, and

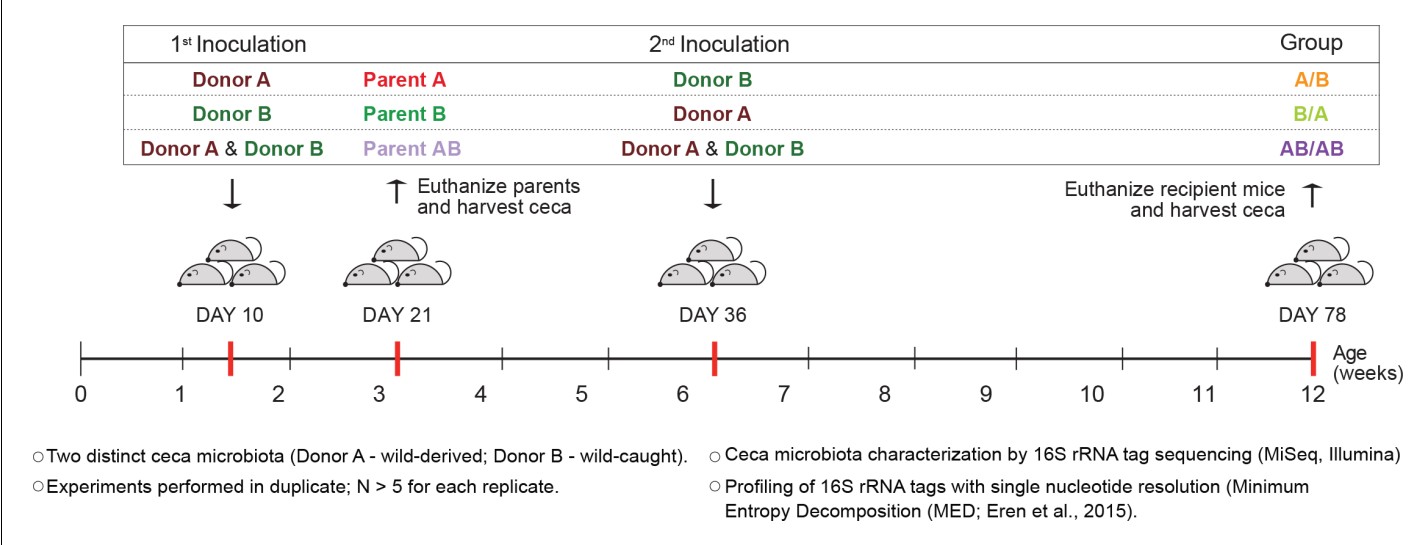

**Figure 1.** Experimental design to elucidate the importance of colonization history in ex-germ-free mice colonized with whole donor microbiomes. Two distinct cecal microbial communities (Donors A and B) that originated from wild or wild-derived *Mus musculus domesticus* and were standardized in laboratory mice (see *Figure 1—figure supplement 1*) were inoculated into germ-free C57BL/6 mice in different order, A/B (A inoculated at day 10 ± 2, and B at day 36 ± 2), B/A (B inoculated at day 10 ± 2, and A at day 36 ± 2), and AB/AB (A and B inoculated together at both days 10 ± 2 and 36 ± 2). Cecal microbiota was characterized in samples collected in the former pubs at day 78 ± 2 and parent mice at day 21 ± 1 by 16S rRNA gene sequencing. Each experiment was done in duplicate (N ≥ 6 for each replicate).

DOI: https://doi.org/10.7554/eLife.36521.003

The following figure supplement is available for figure 1:

**Figure supplement 1.** Standardization of donors' microbiota in laboratory mice.

DOI: https://doi.org/10.7554/eLife.36521.004

Parents AB), while offspring mice were separated by sex. The second inoculation was done at day 36 ± 2 after birth via gavage (*Figure 1*). The cecal microbiome of the recipient mice was characterized at day 78 ± 2, by MiSeq Illumina sequencing of 16S rRNA tags using Minimum Entropy Decomposition (MED) analysis, which allows the differentiation of sequences with only one nucleotide dissimilarity (*Eren et al., 2015*), that are referred to as 'types'.

The rationale to use whole complex microbiomes in contrast to defined simplified communities of strains (which are commonly used in the field of microbial ecology research [*De Roy et al., 2014*]) was to assure that niches were for the most part filled, establishing the competitive interactions present in natural populations. To avoid microbiomes of mice that originate from commercial facilities, which are often derived from gnotobiotic mice and are frequently functionally aberrant and low in diversity (*Lagkouvardos et al., 2016*; *Kreisinger et al., 2014*; *Rosshart et al., 2017*), selected donor mice were caught in the wild or derived from wild mice. One donor mouse was selected from a colony derived from mice caught in the Massif Central region of France and subsequently housed in a laboratory for three generations (*Wang et al., 2014*) (Donor A; referred to as 'wild-derived'). The second donor was a wild mouse caught in Scotland (*Weldon et al., 2015*) (Donor B; referred to as 'wild-caught'). These allopatric mouse populations differed markedly in gut microbiota composition (*Lagkouvardos et al., 2016*). Prior to the actual experiments, the microbial communities from donors A and B were established in germ-free C57BL/6 mice in our facility to standardize the inocula. Cecal bacterial diversity became only marginally reduced in recipient mice 4 weeks after colonization during the standardization when compared to the wild donors, and the majority (85% for donor A and 82% of donor B) of the bacterial types could be detected (*Figure 1—figure supplement 1A*). In addition, an analysis of Bray-Curtis dissimilarities revealed that recipient microbiota clustered tightly with that of the donor (*Figure 1—figure supplement 1B*). Overall, these analyses showed that a large proportion of the microbiota from wild mice can be established in germ-free mice under laboratory conditions with similar taxonomic distributions and proportions.

The experimental model has several unique advantages in that it recapitulates 'real life' conditions during early-life microbiota assembly while keeping ecological variables strictly controlled. The first colonization was done in 10-day-old germ-free mice, which resemble newborn mice as they are devoid of microbes and not yet fully developed, allowing the study of gut microbiota assembly during a time window critical for both microbiota assembly and host postnatal development (*Knoop et al., 2017*). The parent mice become hereby fully colonized, which allows vertical and horizontal transmission of bacteria (i.e. through coprophagia) to the offspring mice up until weaning when they become fully mature. From an ecological perspective, the model is uniquely suitable to disentangle the relative importance of historical processes in community assembly. First, community assembly at local scales (mice) occurs under standardized conditions that included the use of inbred mice under the same husbandry regimen, thereby avoiding variation in deterministic (niche-related) ecological factors. Second, although complex undefined microbiomes were used, the regional species pool, which can profoundly influence the ecological processes that determine community structure at local scales (*Chase and Myers, 2011*), including priority effects (*Sprockett et al., 2018*), is standardized (and to a large degree defined through sequencing). This was achieved by the administration of aliquots of the same inoculum to mice via gavage (a highly efficient method of transmission) and the maintenance of mice in isolators that prevented acquisition of additional members. Consequently, colonization order is the only experimental variable in the model.

## Gut microbiota of recipient mice shows higher diversity when compared to donors

Analysis of donor and recipient microbiota showed that 99.5% of the bacterial types found in the donor microbiomes were also found in the recipient mice, while 93.4% of the types (adding up to 99.3% of the total sequences) in the recipient mice were also detected in the donors. These findings indicate an almost complete transfer of the communities from donors to recipients without an expansion of minority members of the donor communities in the recipient mice. In addition, the abundance distribution of bacterial types in the recipient mice reflected to a large degree those of the donors, showing that transfer of the microbiota did not result in major rearrangements in community structure (*Figure 2A*).

Alpha-diversity of the cecal bacterial community at day 78 was not influenced by colonization order (*Figure 2B*). Interestingly, α-diversity analyses revealed that the microbiomes of recipient mice had a significantly higher number of observed bacterial types compared to the donor communities (*Figure 2B*), irrespective of the order in which the donor microbiomes were introduced ($p=2\times10^{-16}$). Higher diversity was observed when species abundance and evenness (Shannon Index) were considered ($p=1.13\times10^{-4}$) (*Figure 2C*). These results suggest that the niche space in the cecal bacterial community, even in wild mice, is not filled, which agrees with the propositions that most ecosystems in nature are unsaturated (*Cornell and Lawton, 1992*; *Pinto-Sánchez et al., 2014*). This finding is interesting as it provides a potential explanation for why donor-specific bacterial types can colonize the gut of a recipient host with a 'largely' undisrupted microbiota (i.e. patients with metabolic syndrome) after fecal microbiota transplantation (*Li et al., 2016*).

## Early arrival increases colonization success of member of the Donor B community

Hierarchical clustering performed with all bacterial types detected in the donor and recipient mice revealed clustering of the communities dependent on colonization order (*Figure 2A*), indicative of historical contingency. Since it was not possible to analyze priority effects for bacterial types shared among Donors A and B, downstream analyses were performed with bacterial types specific to either donor (Figure 2D and E). A total of 395 bacterial types were detected across the two donor communities, of which 101 were uniquely detected in community A (donor and parents A), 83 were exclusively detected in community B (donor and parents B), and 211 were common to both. Arrival order did not influence the number of successful colonizers unique to donor A (88 ± 6 in A/B, 86 ± 8 in B/A, and 84 ± 9 in AB/AB, $F = 2.75$; p=0.0521), but influenced colonization success of bacterial types unique to donor B (67 ± 8 in B/A, compared to 48 ± 10 in A/B, and 60 ± 7 in AB/AB, $F = 26.5$; $p=8.66\times10^{-11}$). These findings suggest that although a subset of members of the wild-caught B

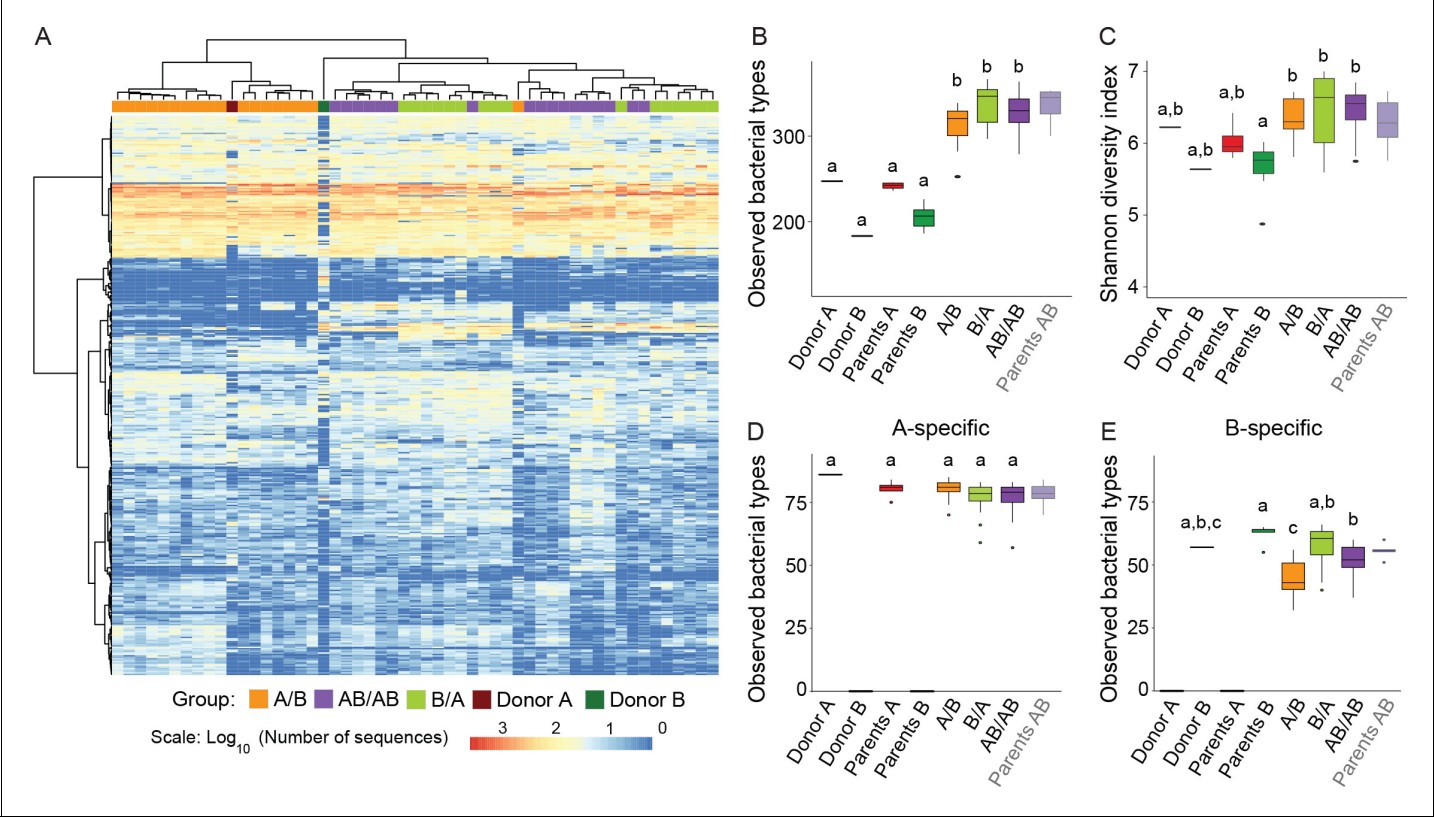

**Figure 2.** Cecal microbiota composition and diversity in donor and recipient mice in relation to colonization order. (A) Heatmap of bacterial types in donor communities A and B and recipient A/B, B/A, and AB/AB mice; bacterial types (rows) and individual mice (columns) were clustered with the Ward's method. (B) Number of bacterial types in and (C) Shannon diversity index of the cecal bacterial communities in donor, parent, and recipient mice. (D and E) Number of bacterial types detected in recipient mice that were unique (specific) to donor c communities A and B . Each experiment was done in duplicate, Donor A (n = 1), Donor B (n = 1), parents A (n = 4), parents B (n = 8), parents AB (n = 6), recipient mice A/B (n = 18), B/A (n = 16), AB/AB (n = 17). In panels B-E, boxplots (ggplot in R) and results from ANOVA statistical analysis are shown (experimental groups that share the same letters are not significantly different, while those indicated with different letters are significantly different) (p<0.05). Parents AB are shown graphically as a reference but were not included in the statistical analysis.
DOI: https://doi.org/10.7554/eLife.36521.005

community are less fit than those of the wild-derived A community (perhaps because the latter are pre-adapted to facility conditions), early arrival can partly offset this fitness disadvantage.

## Assessing the importance of colonization history and dispersal limitation towards shaping gut microbiota structure

Communities at local scales assemble through a combination of niche-related, neutral, and historical processes (*Cavender-Bares et al., 2009*). Since the analysis of β-diversity can provide insight into the importance of ecological processes (*Chase and Myers, 2011*), analyses of Bray Curtis and binary Jaccard dissimilarities were done to determine whether arrival order affected community structure, and how microbiome assembly was impacted by dispersal limitation (as described above, inter-individual niche-related differences should be marginal in our model). The overall cecal microbiomes at day 78 clustered closer to donor A when compared to donor B (*Figure 3A*), supporting the conclusion above that members of the donor A population are on average fitter. This analysis further revealed that communities from A/B mice clustered closer to Donor A when compared to AB/AB and B/A mice, while those of B/A mice clustered closer to Donor B as compared to AB/AB and A/B mice (*Figure 3B*), suggesting that the communities of the recipients are more similar to the donor community that arrived first. Analysis of variance of the bacterial communities of just the recipient mice using Adonis revealed that bacterial profiles (clustered on the basis of Bray-Curtis) were significantly affected by colonization order (all treatments p<0.001, $R^2$ = 0.21; A/B vs B/A p<0.001,

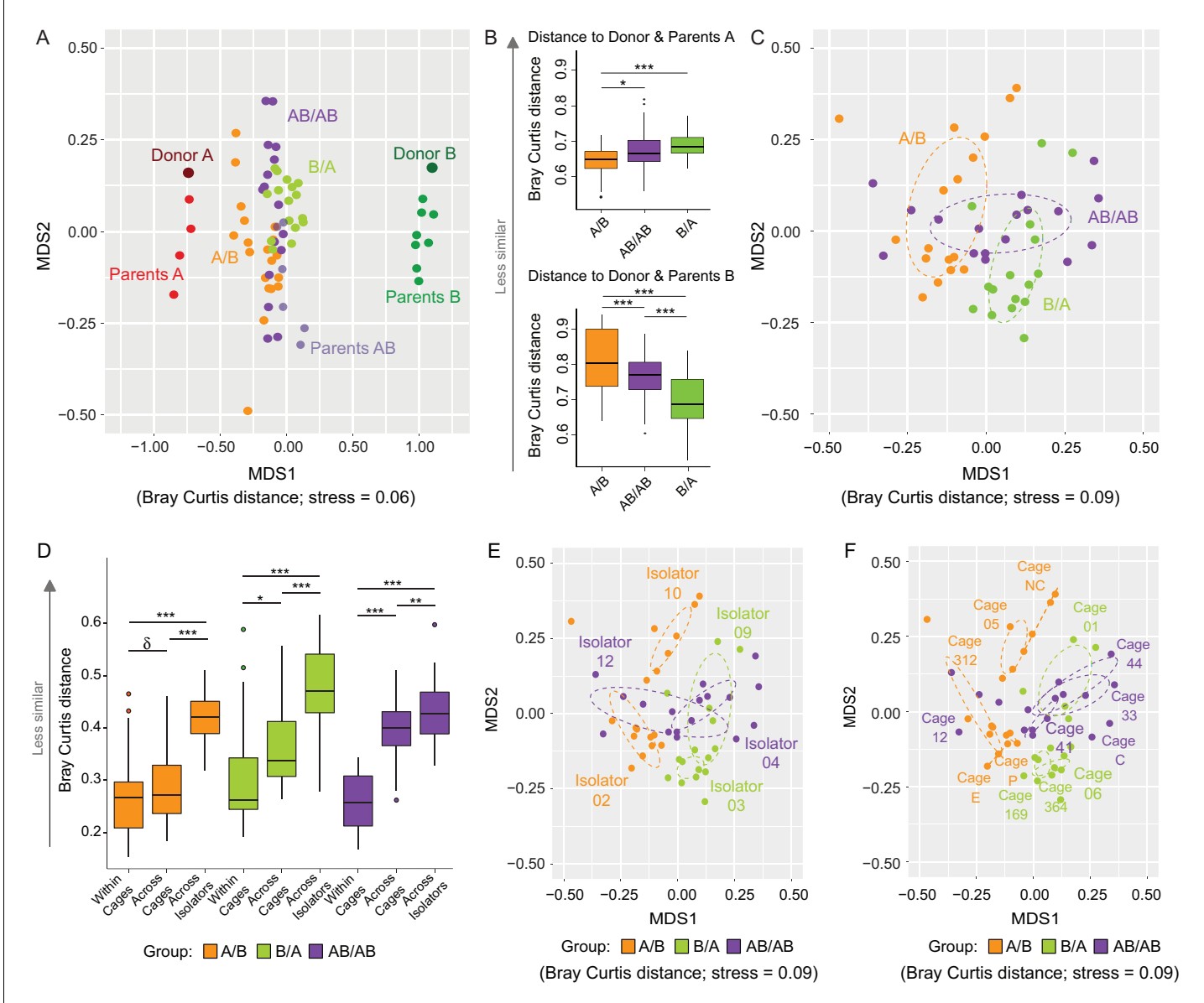

**Figure 3.** Effects of colonization order and dispersal limitations on the overall composition of the cecal microbiota. (**A**) Non-metric multidimensional scaling (NMDS) plots based on Bray-Curtis dissimilarities of donors A and B, parents A and B, and recipient A/B, B/A, and AB/AB mice. (**B**) Average Bray Curtis dissimilarities between cecal bacterial communities in recipient mice with those of the donor and parent mice. (**C**) NMDS plots based on Bray-Curtis dissimilarities of recipient A/B, B/A, and AB/AB mice. (**D**) Average Bray-Curtis distances of the cecal microbiota profile of mice housed within the same cage, across cages within an isolator, and across isolators; statistical tests were performed within treatments to exclude the effect of colonization order. (**E and F**) NMDS plots (Bray-Curtis dissimilarity) with samples color-coded by treatment and grouped by the isolator or cages they were housed in. Each experiment was done twice, Donor A (n = 1), Donor B (n = 1), parents A (n = 4), parents B (n = 8), parents AB (n = 6), recipient mice A/B (n = 18), B/A (n = 16), AB/AB (n = 17), isolator 02 (n = 11), isolator 03 (n = 9), isolator 04 (n = 11), isolator 09 (n = 7), isolator 10 (n = 7), isolator 12 (n = 6), cages (n = 2–5). B and D show boxplots (ggplot in R), and the data was analyzed with ANOVA; ***p<0.001, **p<0.01, *p<0.05, δp<0.1.
DOI: https://doi.org/10.7554/eLife.36521.006

The following figure supplements are available for figure 3:

**Figure supplement 1.** Determination of the importance of colonization history in mouse experiments using whole donor microbiomes.
DOI: https://doi.org/10.7554/eLife.36521.007

**Figure supplement 2.** Impact of dispersal limitation on gut microbiome composition.
DOI: https://doi.org/10.7554/eLife.36521.008

$R^2$ = 0.23; A/B vs AB/AB p<0.001, $R^2$ = 0.16; and B/A vs AB/AB p=0.002, $R^2$ = 0.11 (*Figure 3C*). Binary Jaccard-based analysis confirmed these differences (*Figure 3—figure supplement 1*), indicating a measurable influence of colonization order on gut microbiota β-diversity.

The experiment further demonstrated a clear importance of dispersal limitation in gut microbiota assembly. Cecal microbiomes of mice housed within the same cage were more similar (less β-diversity) than mice housed in different cages within the same isolator (p=0.092 within A/B; p=0.019 within B/A; and p<1.00×10$^{-7}$ within AB/AB; *Figure 3D*), while mice housed in different isolators (no opportunity for dispersal) had the largest compositional differences (p<1.00×10$^{-7}$ within A/B and B/A; p=6.1×10$^{-3}$ for AB/AB; *Figure 3D*). β-diversity analysis based on Bray Curtis dissimilarities revealed significant clustering by both isolator (*Figure 3E*) and cage (*Figure 3F*). Adonis analysis performed using 'cage', 'isolator', and 'colonization order' as explanatory variables revealed that the data was best explained by the factor cage (p<0.001, $R^2$ = 0.66), followed by isolator (p<0.001, $R^2$ = 0.46) and colonization order (p<0.001, $R^2$ = 0.21). Analogous results were obtained when using binary (presence/absence) measurements (*Figure 3—figure supplement 2*). These results demonstrate that dispersal limitation can cause cage effects even if individual animals acquire exactly the same species members in a completely standardized environment. This finding has substantial implications for the design and interpretation of mouse experiments, as it suggests that a difference among mouse groups can arise not only from familial transmission (*Ubeda et al., 2012*) but also solely from ecological drift. However, although dispersal limitation does explain the largest proportion of β-diversity in the cecal bacterial population of recipient mice, colonization order makes a clear measurable contribution.

## Bacterial types affected by colonization order and mechanisms of priority effects

The impact of arrival order on microbiota assembly prompted the evaluation of specific taxonomic groups affected, using linear mixed models. No significant differences in phyla, families, or genera abundances were observed across experimental groups (data not shown). However, 20 bacterial types were significantly influenced by colonization order (*Supplementary file 1*), indicative of priority effects. The taxonomic distribution of these bacterial types mirrored the overall community profile, showing that no particular phylum was specifically more affected by colonization history. The fact that colonization order impacts bacterial types but not higher taxonomic levels points to the importance of competitive interactions as a major mechanism driving the observations.

To gain insight into the mechanisms by which priority effects emerge, it was assessed whether bacterial types affected by colonization order were overrepresented when inoculated first or overrepresented when introduced second. If a bacterial type is overrepresented when inoculated first, it would occur either through niche-preemption, monopolization (the evolutionary process by which early arrivals to a new patch adapt to the local conditions, gaining an advantage over later colonists) or niche-modification that provides an advantage to the earlier colonizer, all of which are inhibitory (*Fukami, 2015*). In contrast, an overrepresentation of a bacterial type that that arrives second can occur through facilitative niche-modification (*Fukami, 2015*). This analysis revealed that priority effects were mostly inhibitory (*Supplementary file 1*), as in 90% of the cases, bacterial types were more successful when introduced in the first inoculation (two examples, Types 4996 and 0857, are shown in *Figure 4A*). However, instances of facilitative effects were also observed as two bacterial types (Firmicutes) had a higher abundance when introduced second (Type_4501 is shown as an example in *Figure 4A*; *Supplementary file 1*).

Inhibitory effects can occur through niche pre-emption, where early arriving species inhibit the colonization of those arriving later through competitive exclusion and limiting similarity (*Fukami, 2015*). As closely related organisms often display overlapping niches, niche pre-emption might manifest by related organisms to exclude one another (*Cavender-Bares et al., 2009*; *Violle et al., 2011*; *Fukami et al., 2016*). However, colonization success of bacterial types (evaluated as the fraction of mice colonized or the average abundance of the bacterial type) was not a function of phylogenetic distance to members of the opposite donor community (*Figure 4—figure supplement 1*). This finding suggests that bacterial functions among gut microbiota members are not necessarily related to phylogeny. Although an overall phylogenetic signal could not be detected, there were four instances in which closely related bacterial types introduced early prevented the establishment of related bacterial types that arrived later (1 to 7 mismatches in 16S rRNA gene tag, >97% –

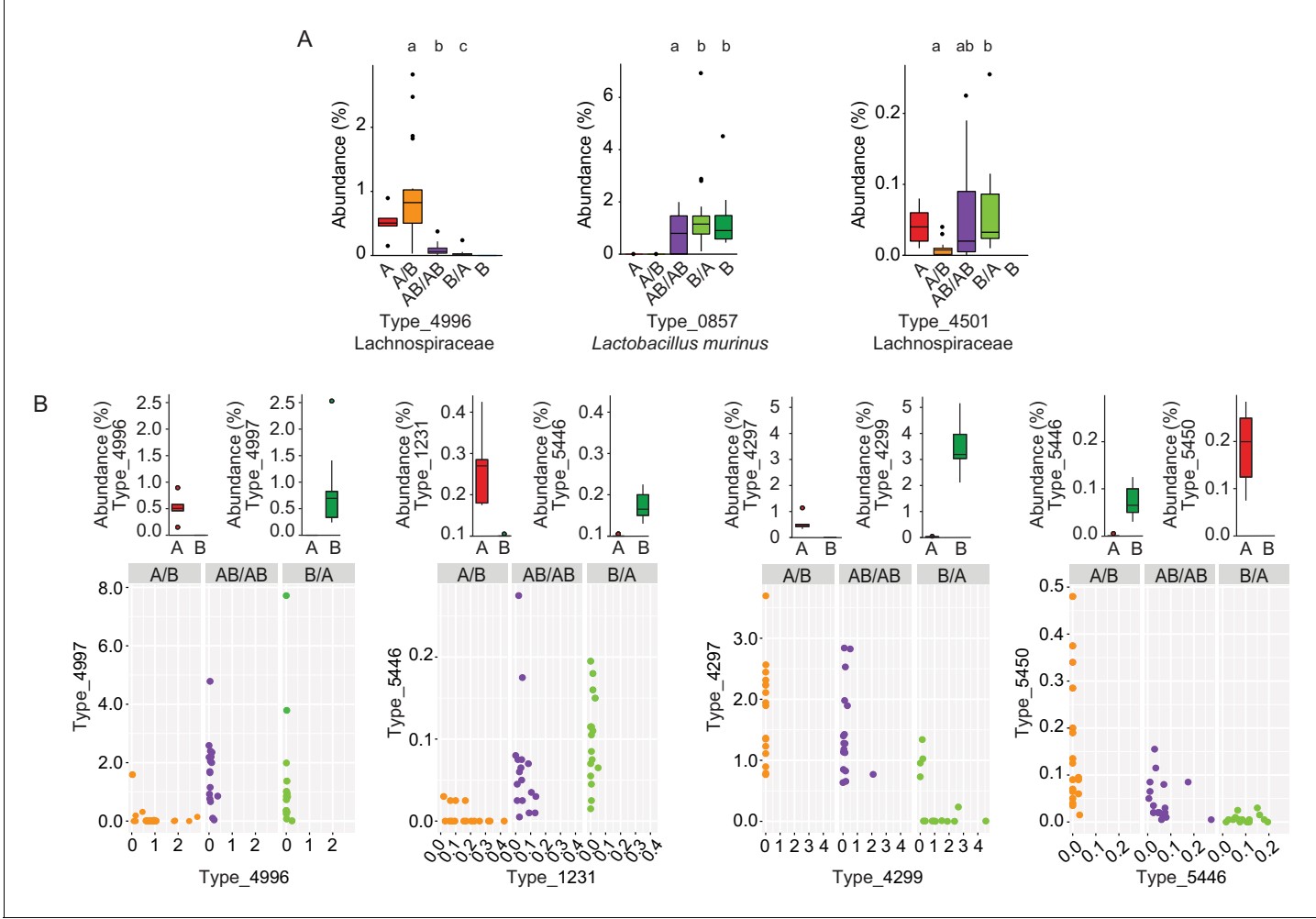

**Figure 4.** Bacterial types affected by colonization order and mechanisms of priority effects. (**A**) Examples of bacterial types with an advantage when inoculated first (Type_4996 and Type_0857), indicative of inhibitory priority effect, and when inoculated second (Type_4501), indicative of facilitation. Different letters indicate statistically significant differences between groups (values of donors A and B were not included in the statistical analysis but were plotted as a reference). (**B**) Associations between bacterial types affected by colonization order with phylogenetically related types in the competing community. Type_4996 and Type_4997 (7 mismatches in 16S rRNA gene tag, >97% identity), Types_4297 and Types_4999 (1 mismatch in 16S rRNA gene tag, >99% identity), Type_5446 and Type_1231 (4 mismatches in 16S rRNA gene tag, >98% identity), and Type_5450 and Type_5446 (2 mismatches in 16S rRNA gene tag, >99% identity) show negative correlations, indicative of niche pre-emption. Each experiment was done in duplicate, Donor A (n = 1), Donor B (n = 1), recipient mice A/B (n = 18), B/A (n = 16), AB/AB (n = 17). In panel (**A**) and top graphs in panel (**B**), boxplots are generated in ggplot in R. In panel (**A**) results from ANOVA statistical analysis are shown (experimental groups that share the same letters above are not significantly different, while those indicated with different letters are significantly different)(p<0.05).
DOI: https://doi.org/10.7554/eLife.36521.009

The following figure supplement is available for figure 4:

**Figure supplement 1.** Correlations between A-specific and B-specific types in donor communities and the closest related types in the competing community, in the different groups of mice (A/B, B/A, and AB/AB).
DOI: https://doi.org/10.7554/eLife.36521.010

>99% similarity; **Figure 4B**). In two of the cases (Type_5446 vs. Type_1231, and Type_5450 vs. Type_5446) the two types co-existed when introduced together in group AB/AB, and earlier colonization makes either of them more competitive (higher relative abundance) (**Figure 4B**). In the other two cases (Type_4996 vs. Type_4997, and Type_4297 vs. Type_4299), one type excludes the other type when introduced together, but earlier colonization can overcome this fitness difference and give an advantage to the type that displays a lower degree of fitness (**Figure 4B**). Overall, these

patterns provide evidence for the importance of priority effects that are inhibitory through niche pre-emption.

## Timing of arrival impacts persistence of individual colonizers

The findings described above established the importance of colonization order in community assembly when whole microbiomes were acquired in different succession. These experiments raise the question on the exact contribution of specific members within these communities toward historical contingency, and how they themselves are impacted by arrival timing. Thus, in a separate set of experiments, four bacterial strains of mouse origin (*Bacteroides vulgatus* RJ2H1, *Lactobacillus reuteri* Lpuph1, *Lactobacillus johnsonii* DPPM, and *Clostridium cocleatum* ATCC 29902$^T$) were gavaged together into germ-free mice either before (at 5 days of age), soon after (at 14 days of age), or more than 3 weeks after (at 36 days of age) the mice received a complex cecal microbiota (obtained from pooled cecal samples of five C3H/HeN mice raised in conventional housing at the University of Nebraska Gnotobiotic Mouse Facility) at 10 days of age (*Figure 5A*). Using strain-specific quantitative PCR (qPCR), abundance of each strain was quantified in fecal samples collectedly weekly throughout the 36 days period after the respective inoculations, as well as in the ceca collected at day 78.

As shown in *Figure 5B*, arrival timing significantly impacted colonization patterns for two of the strains. Cell numbers of *L. reuteri* Lpuph1 were significantly higher (p=8.99×10$^{-8}$ time*group interaction) at all time points when introduced on day 5 compared to days 14 and 36 (*Figure 5B*), and the strain was only detected at the end of the experiment when it was introduced on day 5 but not at later time points (detectable in two mice, 1.71 × 10$^6$ cells/g of cecal content). For *C. cocleatum* ATCC 29902$^T$, the timing of immigration was absolutely crucial, as the organism was only detected in mice when the strain was introduced at day 5 (p<2.20×10$^{-16}$ time*group interaction) (*Figure 5B*). For *B. vulgatus* RJ2H1 and *L. johnsonii* DPPM, time of inoculation did not affect colonization success, and both strains were able to stably colonize mice at equivalent levels independently of arrival time (*Figure 5B*). These findings indicate that specific microbiota members can differ substantially in the degree to which arrival timing relative to competitors influences their colonization success.

Theory predicts that functional similarity (niche overlap) among colonists makes the outcome of interspecific competition sensitive to arrival order (*Fukami, 2015*; *Sprockett et al., 2018*). This prediction was tested by determining the closest relative to the introduced species within the donor microbiome from the 16S rRNA data. The analysis showed that the two strains in the cocktail that were favored by early arrival, *L. reuteri* Lpuph1 and *C. cocleatum* ATCC 29902$^T$, were 100% identical in their 16S rRNA gene sequence to bacteria present in the donor community, Type_3050 (2.15 ± 2.60% abundance) and Type_3983 (0.42 ± 0.15% abundance) (*Figure 5C*). In contrast, the two stable colonizers not impacted to arrival order had no closely related organism in the donor community (*Figure 5C*). Although other mechanisms cannot be excluded, our findings therefore suggest that early arrival of a bacterium is critical for its ecological success if the community contains members that overlap with its niche. If such competitors are absent, then stable colonization is independent of timing and can occur later in life, equivalent to findings in adult humans in a recent study with *Bifidobacterium longum* ssp. *longum* (*Maldonado-Gómez et al., 2016*).

## Differences in arrival timing of the four-strain cocktail cause historical contingency in gut microbiota assembly

To test if differences in arrival time of a small number of early colonizers can impact gut microbiota assembly, the cecal bacterial community of the 78-day-old mice that received the four-strain cocktail at different time points was analyzed. No differences in α-diversity were detected across treatments (number of observed bacterial types p=0.464, and Shannon Index p=0.841; *Figure 5D*). However, structure of the fecal bacterial communities did show significant differences based on when the cocktail of specific colonizers was introduced (Adonis; Bray Curtis p=0.041, $R^2$ = 0.08; Jaccard Index p=0.013, $R^2$ = 0.08). The fecal microbiota in mice inoculated with the four-strain mix at day 5 differed in their overall composition at day 78 compared to those that received the cocktail of strains at day 14 (Adonis; Bray Curtis p=0.047, $R^2$ = 0.07; Jaccard Index p=0.027, $R^2$ = 0.06). There was no difference between mice colonized with the cocktail at day 14 and day 36 (Adonis; Bray Curtis p=0.114, $R^2$ = 0.06; Jaccard Index p=0.073, $R^2$ = 0.06) (*Figure 5E*). Twenty-six bacterial types

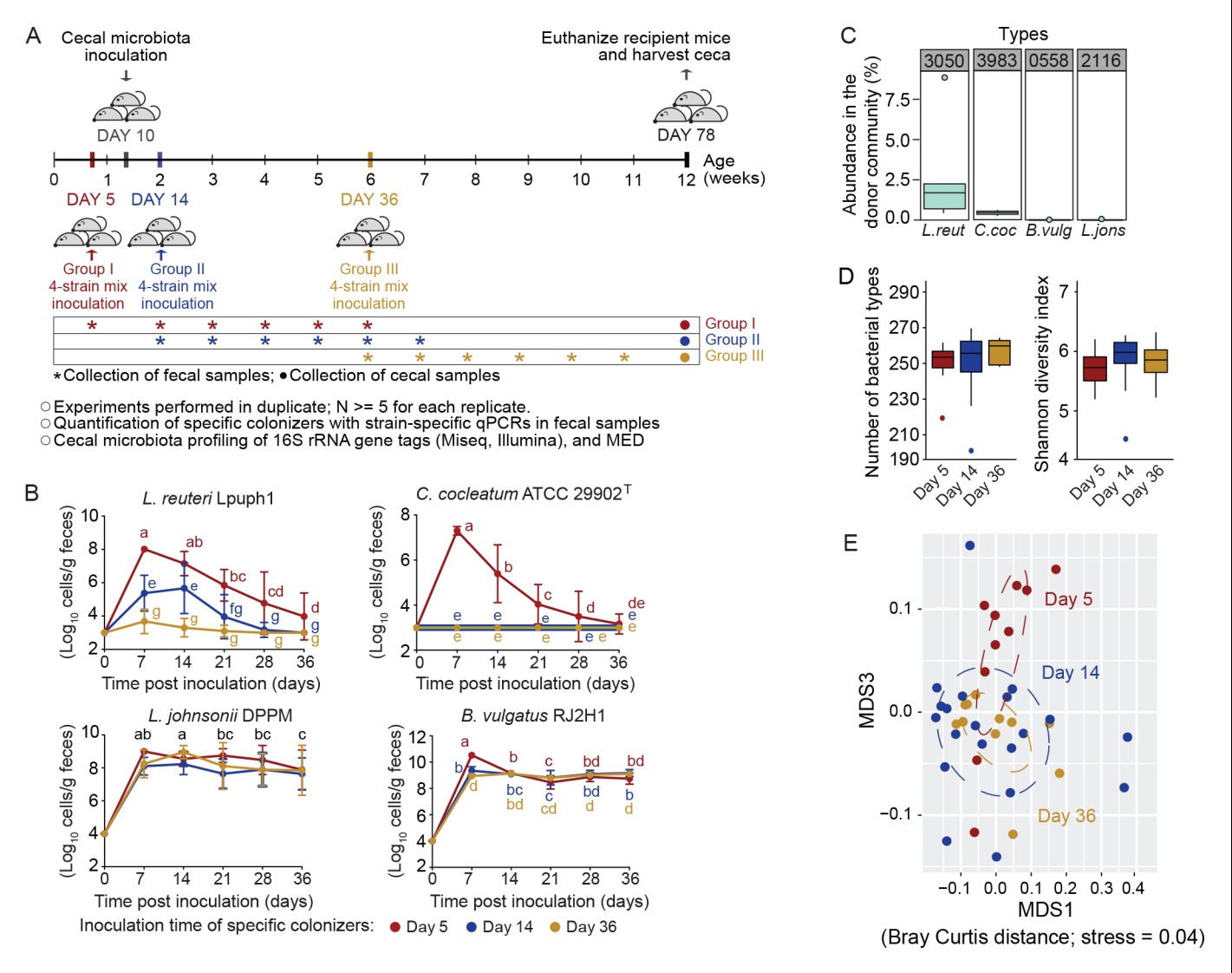

**Figure 5.** Arrival timing of specific colonists impact their own colonization success and the trajectory of cecal microbiota assembly. (**A**) Experimental design. *Lactobacillus reuteri* Lpuph1, *Lactobacillus johnsonii* DPPM, *Bacteroides vulgatus* RJ2H1, and *Clostridium cocleatum* 29902[T], were inoculated into mice at days 5 (n = 10), 14 (n = 22) and 36 ± 2 (n = 11) after birth, while a cecal microbial community (derived from C3H/HeN mice) was introduced at day 10 ± 1. Fecal samples were collected pre-treatment on the day of inoculation of the strain cocktail and weekly throughout 5 weeks post-inoculation. At day 78 ± 2 mice were euthanized and their ceca harvested for taxonomic characterization with MiSeq (Illumina) technology. (**B**) Cell numbers of the specific colonizers (determined by strain-specific qPCR) in fecal samples collected throughout a 5-week period following inoculation. (**C**) Abundance of bacterial types closest related to the specific colonizers in the donor cecal community based on 16S rRNA gene tag data. Type_3983 had 2 mismatches to *C. cocleatum* 29902[T], while Type_3050 was identical to *L. reuteri* Lpuph1. The closest match to *B. vulgatus* RJ2H1 was Type_0558 and had 8 mismatches, while the closest bacterial type to *L. johnsonii* DPPM was Type_2116 with 15 mismatches. (**D**) Number of bacterial types and Shannon diversity index of the cecal microbiota in mice inoculated with the specific colonizers at days 5, 14 and 36 at age 78 days. E, NMDS plot (based on Bray-Curtis dissimilarity) showing cecal bacterial community at age 78 days in mice that received the four specific colonists at 5, 14, and 36 days of age. In panel (**B**) mean ±standard derivation of two replicates per experiment, and data were analyzed by ANOVA with interaction of time and treatment evaluated; different letters across time points or experimental groups represent significantly different values (p<0.05). In panels (**C and D**) boxplots are shown (ggplot in R). In panel D, ANOVA analysis was performed; results with no significant differences across treatments detected.
DOI: https://doi.org/10.7554/eLife.36521.011

significantly differed in abundance across treatments (random forest analysis; mean importance >2; *Supplementary file 2*), none of which represented the strains from the cocktail. Overall, these findings demonstrate that differences in arrival timing of just a small number of early colonizers can bring about marked changes in the trajectory of gut microbiota assembly.

## The adaptive immune system is not a major contributor to historical contingency in gut microbiota assembly

From a microbial ecosystem perspective, the vertebrate gut is unique in that not only can the microbial community members adapt (through monopolization) during assembly (*De Meester et al., 2016*), but the habitat (host) can also respond and adapt towards the community during postnatal development (*Atarashi et al., 2011*; *Ivanov et al., 2009*). Specialized features of the adaptive immune system allow the host to develop immunological tolerance to gut bacteria, for example through antigen-specific regulatory T cells (*Belkaid and Hand, 2014*; *Cebula et al., 2013*). These immune processes differ between the neonatal and adult immune system (*Sudo et al., 1997*; *Olszak et al., 2012*) such that early host exposure to specific bacteria may precondition the immune environment for bacterial colonization (*Knoop et al., 2017*; *Gensollen et al., 2016*). Differences in bacterial arrival time during community assembly could therefore cause historical contingency through adaptive immune responses that favor early colonizers during a 'window of opportunity' (*Knoop et al., 2017*; *Torow and Hornef, 2017*; *Koskella et al., 2017*). To test whether adaptive immune responses during postnatal immune development contribute to historical contingency in gut microbiota assembly, the same set of experiments as described above was repeated in mice of the same genetic background (C57BL/6) but without adaptive immunity (*Rag1$^{-/-}$*), using the same donor microbial communities and four-strain cocktail, and compared the finding with wild-type (WT) mice.

Equivalent to the findings in WT mice, the introduction of two whole cecal bacterial communities into *Rag1$^{-/-}$* mice led to a significant increase in species richness (p=6.63×10$^{-14}$) when compared to the individual donor communities, regardless of colonization order (*Figure 6A*). However, when compared to WT mice, *Rag1$^{-/-}$* mice in two treatments (B/A and AB/AB) had a slight but significantly lower number of observed bacterial types (p=7.3×10$^{-4}$ and p=0.028, respectively) (*Figure 6B*), and one treatment (B/A) had a lower Shannon diversity index (p=0.043; data not shown). In addition, the overall cecal microbiota differed between WT and *Rag1$^{-/-}$* mice (Adonis based on Bray-Curtis dissimilarities, p<0.001, $R^2$ = 0.07) (*Figure 6C*), with 21 bacterial types to significantly differ in abundance across genetic backgrounds (*Supplementary file 3*). These results suggest that the adaptive immune system increases diversity and affects colonization of specific bacterial strains, although the findings contrast with previous research that showed that diversity between WT and *Rag1$^{-/-}$* mice and Zebrafish were not different (*Zhang et al., 2015*; *Stagaman et al., 2017*).

Colonization history remained important in *Rag1$^{-/-}$* mice, but some differences were detected. The impact of early colonization on the similarity of recipient communities to the donor communities was reduced in *Rag1$^{-/-}$* mice (*Figure 6D*), with only the B/A mice being significantly more similar to donor/parents B when compared to A/B mice (*Figure 6E*). In addition, the enhanced level of colonization of donor B-specific bacteria when introduced early observed in WT mice (*Figure 2E*), was not detected in *Rag1$^{-/-}$* mice, and the average number of donor-B-specific bacterial types successfully colonizing the gut did not differ across treatments (p=0.166, p=0.279, and p=0.980) (data not shown). These results illustrate that the adaptive immune system is a contributor to the successful establishment of specific early arriving bacterial types in the gut. However, just as in WT mice (*Figure 3C*), there was significant clustering of the ceca microbiota by colonization order in *Rag1$^{-/-}$* mice (Adonis; Bray Curtis and Jaccard dissimilarity, p<0.001, $R^2$ = 0.25; p<0.001, $R^2$ = 0.22, respectively) (*Figure 6F*). β-diversity was impacted to a larger degree by colonization order than by the adaptive immune system (*envfit* analysis of correlation between MDS axes and variables; p=0.001; $R^2$ = 0.15, and $R^2$ = 0.07, respectively), indicating a stronger effects of colonization history over host adaptive immunity.

To compare which bacterial types were affected, differences in bacterial abundance were analyzed in WT and *Rag1$^{-/-}$* mice using a linear mixed model with genetic background and colonization order as the main effects, isolator as a random effect, and an interaction term between colonization order and genetic background (and FDR correction was applied to the *P values*). Together, forty-three bacterial types were significantly affected by colonization order, with a majority (81%) showing patterns indicative of inhibitory priority effects (*Supplementary file 4*). Importantly, those included 19 out of the 20 bacterial types previously identified to be significantly affected in WT mice, while only two bacterial types showed a significant interaction between genetic background (WT vs *Rag1$^{-/-}$*) and colonization order (data not shown), illustrating similar ecological effects in the two genetic backgrounds. Overall, these findings revealed that colonization history remained an

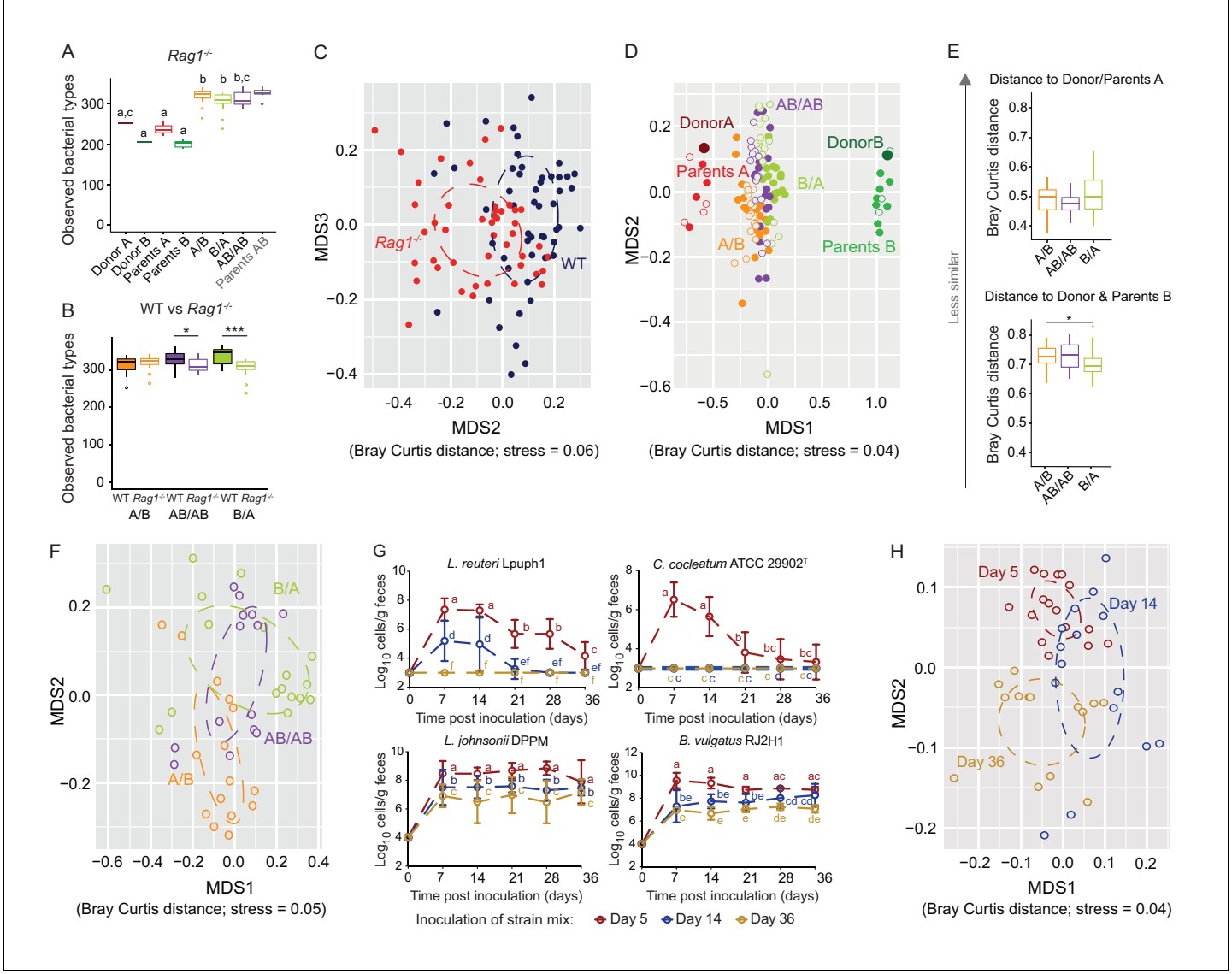

**Figure 6.** Determination of the role of colonization history in *Rag1*⁻/⁻ mice. (**A**) Boxplots showing the number of bacterial types in the donor A and B communities (shown are the donor mice; n = 1 each), the parents inoculated with A (n = 4) or B (n = 3), and recipient mice colonized with the donor communities in different order (A/B; n = 15), B/A; n = 17), and AB/AB; n = 14). (**B**) Boxplots of bacterial types in WT versus *Rag1*⁻/⁻ mice. (**C**) NMDS plots based on Bray-Curtis dissimilarities of cecal microbiota of all experimental groups in WT (blue) and *Rag1*⁻/⁻ (red) mice (n = 51, and n = 46, respectively). (**D**) NMDS plot based on Bray-Curtis dissimilarities in recipient WT (filled symbols) and *Rag1*⁻/⁻ (open symbols) mice that received treatments A/B (n = 18 WT; n = 15 *Rag1*⁻/⁻), B/A (n = 16 WT; n = 17 *Rag1*⁻/⁻), and AB/AB (n = 17 WT; n = 14 *Rag1*⁻/⁻) together with donors A (n = 1) and B (n = 1) and parents A (n = 4 WT; n = 4 *Rag1*⁻/⁻) and B (n = 8 WT; n = 3 *Rag1*⁻/⁻). (**E**) Average Bray Curtis dissimilarities between cecal bacterial communities in recipient *Rag1*⁻/⁻ mice with those of the donor and parent mice. (**F**), NMDS plots based on Bray-Curtis dissimilarities of cecal in *Rag1*⁻/⁻ mice that received the A/B (n = 15), B/A (n = 17), and AB/AB (n = 14) treatments. (**G**) Absolute cell numbers of *Lactobacillus reuteri* Lpuph1, *Lactobacillus johnsonii* DPPM, *Bacteroides vulgatus* RJ2H1, and *Clostridium cocleatum* 29902ᵀ in fecal samples of *Rag1*⁻/⁻ mice when inoculated into mice at days 5 (n = 16), 14 (n = 15) and 36 (n = 14)±2 days after birth, while a complex microbiota was established at day 10. (**H**), NMDS plot based on Bray-Curtis dissimilarity showing cecal bacterial community at 78 ± 2 days in *Rag1*⁻/⁻ mice that received the four specific colonists at 5 (n = 16), 14 (n = 15) and 36 (n = 14)±2 days after birth. In panels (**A and B**), statistical analyses were done with ANOVA. In panel G, mean ±standard derivation of two replicates per experiment are shown, and data was analyzed by ANOVA with interaction of time and treatment. In panels (**A and G**), different letters across time points or experimental groups represent significantly different values (p<0.05). ***p<0.001, **p<0.01, *p<0.05, δ p<0.1.

DOI: https://doi.org/10.7554/eLife.36521.012

The following figure supplement is available for figure 6:

**Figure supplement 1.** Comparison of fecal cell numbers of specific colonists in WT and *Rag1*-/- mice in relation to arrival timing.
DOI: https://doi.org/10.7554/eLife.36521.013

important contributor to microbiota assembly from two complex donor communities in the absence of adaptive immunity.

Interestingly, experiments that tested the importance of arrival timing of specific colonizers revealed that both priority effects and historical contingency of community assembly were larger in $Rag1^{-/-}$ mice. The colonization patterns of *L. reuteri* Lpuph1 and *C. cocleatum* ATCC 29902$^T$ were not different in $Rag1^{-/-}$ and WT mice (p=0.22), and colonization timing remained important (*Figure 6G* and *Figure 6—figure supplement 1*). In contrast, the population sizes of *L. johnsonii* DPPM and *B. vulgatus* RJ2H1, which were not affected by colonization timing in WT mice, were lower in $Rag1^{-/-}$ mice when colonization occurred later (*Figure 6G*, *Figure 6—figure supplement 1*). For both strains, later colonization (14 and 36 days) resulted in significantly lower cell numbers in $Rag1^{-/-}$ mice compared to mice colonized at day 5 (p=$1.69\times10^{-5}$ for *L. johnsonii*, and p=0.004 for *B. vulgatus*) and to WT mice colonized at days 14 and 36 (p=$1.69\times10^{-5}$ for *L. johnsonii*, and p=$1.33\times10^{-7}$ for *B. vulgatus*).

The impact of the specific colonizers on the historical contingency of the bacterial community was also larger in $Rag1^{-/-}$ mice, as colonization of the strain cocktail at 5 and 14 days resulted in different microbiome configurations when compared to each other and when compared to mice colonized at 36 days (Adonis test across all groups; Bray Curtis dissimilarity p<0.001, $R^2$ = 0.19, Jaccard dissimilarity p<0.001, $R^2$ = 0.17) (*Figure 6H*). Random forest analysis revealed that 46 bacterial types differed in the assembled community (78 days) of $Rag1^{-/-}$ mice inoculated at days 5, 14 or 36 (*Supplementary file 5*), showing that a greater number of types were affected by colonization order when compared to WT mice.

Overall, although the comparisons of WT and $Rag1^{-/-}$ mice indicate that the adaptive immune system can contribute to the successful establishment of specific early colonizers, priority effects and historical contingency still occured in $Rag1^{-/-}$ mice, and both processes were, in several measurements, increased when compared to WT mice. More bacterial types were effected by colonization order in $Rag1^{-/-}$ mice in the experiments with the bacterial cocktails, and effect sizes in Adonis tests that explained the impact of colonization order on community variation were, on average, higher in both experiments. Especially the experiments using the cocktail of specific colonizers clearly showed an enhanced role of historical processes in $Rag1^{-/-}$ mice on both the colonization success of the individual microbes as well as the historical contingency of the entire bacterial community.

## Discussion

Most of the inter-individual variability observed in mammalian gut microbial communities is unaccounted for (*Benson et al., 2010*; *Wang et al., 2016*; *Goodrich et al., 2016*). Previous work aiming to elucidate the factors that drive diversity patterns focused mostly on deterministic processes while neglecting stochastic ecological elements, that according to theory, are implicated in shaping community assembly (*Vellend, 2010*; *Cavender-Bares et al., 2009*; *Sprockett et al., 2018*). By establishing that historical processes can drive β-diversity of the gut microbiota in a setting where host genetics, diet, and the regional species pool are standardized, this study demonstrated that individuality in gut microbiota composition can arise solely from a variation of the timing by which microbes are acquired. Although the relative importance of assembly history appears small in our experiments (and less than that of dispersal limitation), effects of species arrival history in real life are predicted to be amplified over time and space (*Fukami, 2015*). Our finding that differences in arrival timing by only four members by a few days can produce a measurable alteration in the long-term trajectory in the development of the community is therefore especially relevant. Given that there are hundreds if not thousands of species that assemble into gut microbiomes, most of whose acquisition is likely to some degree stochastic, historical events have the potential to add up to contribute substantially to individuality of gut microbiomes. Our findings therefore have important implications on our understanding of how microbiotas assemble. They further illuminate the need to develop statistical models and algorithmic approaches that could be incorporated into existing algorithms for population-driven analyses (e.g. GWAS and MWAS) to accurately measure the relative contributions of ecological factors, host genetic variation, and other deterministic factors in large-scale population-based studies.

Although our findings clearly establish the importance of colonization history in gut microbiota assembly and suggest an importance of inhibitory priority effects, they provided limited insight into

the exact mechanisms that cause historical contingency. Priority effects can arise through niche pre-emption or modification (*Fukami, 2015*; *Sprockett et al., 2018*), or through in situ evolution by which earlier colonizers adapt to gain an advantage over later colonizers through monopolization (*De Meester et al., 2016*). These processes might also be impacted by interactions across domains of life (bacteria, eukaryotes, archaea) and even bacteriophages. Future studies should therefore employ metagenomic sequencing and 'trait-based' approaches with careful consideration of the interactions between the functional, taxonomic, and genetic features of both the colonists and the assembled microbiomes in an attempt to determine the relative importance of niche-preemption, niche-modification (including inter-domain interactions), and monopolization in historical contingency. In addition, arrival timing of microbes might also change the trajectory of gut microbiota assembly by inducing developmental, physiological, and immunological changes in the host that then in themselves shape the microbiota. In other words, the timing of species immigration might not only impact microbe-microbe interactions that would qualify as classical priority effects, but also reciprocal interactions of microbes and the host. Future studies should attempt to determine to what degree the latter contributes to historical contingency in gut microbiota assembly. If relevant, host factors and host-microbe interactions would have to be incorporated into ecological theory in order for its application to be useful for our understanding of host-associated microbial communities.

In this study, the role of a key inducible host factor in historical contingency, the adaptive immune system, was systematically analyzed. We hypothesized, like others (*Sprockett et al., 2018*; *Koskella et al., 2017*), that ecological consequences similar to those of priority effects would arise through early colonizers gaining an advantage through the induction of tolerance by adaptive immune responses (*Knoop et al., 2017*). However, our comparisons of WT and *Rag1*$^{-/-}$ mice indicated that the importance of historical processes was, with few exceptions, not reduced in the absence of adaptive immunity, and nearly all the taxa affected by colonization order in WT mice were also among those affected in *Rag1*$^{-/-}$ mice. Moreover, the overall effect of colonization history was actually amplified in most measurements in *Rag1*$^{-/-}$ mice. A potential explanation for this finding is a higher relative contribution of selection (deterministic processes) in *Rag1*$^{-/-}$ mice compared to their WT counterparts. Ex-germ-free *Rag1*$^{-/-}$ mice develop inflammation in the intestinal epithelial lining in response to bacterial colonization (*Peterson et al., 2007*), and this inflammatory milieu could exert a higher selective pressure on the bacterial community. Stronger selection is predicted to increase competition, which in itself increases the importance of priority effects (*Fukami, 2015*), providing an explanation for the impact of sequence of arrival on the population of *L. johnsonii* and *B. vulgatus* in *Rag1*$^{-/-}$ but not WT mice. In summary, it appears that for some specific species, the adaptive immune system might alter niches, allowing, for example, donor-B-specific lineages to persist (*Figure 2E*), a finding only obtained with WT but not *Rag1*$^{-/-}$ mice. However, most of our observations do not support the hypothesis that interactions between microbes and the adaptive immune system are a major contributor to historical contingency. Instead, our analysis on microbe-microbe associations (*Figure 4*) suggests ecological interactions such as inhibitory priority effects and potentially niche-preemption as the dominant processes. Future studies are necessary to determine if additional inducible host factors (innate immune functions, defensins, mucus, etc.) constitute a mechanism by which arrival timing of microbes causes historical contingency in gut microbiota assembly.

Although our understanding on how ecological theory can be applied to the gut microbiome is still vastly incomplete, our findings contribute to an emerging historical perspective that could explain the spatial and temporal patterns of diversity described for the gut microbiota (*Walter and Ley, 2011*). The seemingly paradoxical characteristics of the gut microbiota, namely resilience and high intra-individual stability despite large inter-individual variability, are consistent with a view in which stochastic historical events such as chance colonization, random extinction, ecological drift, and monopolization, in combination with niche pre-emption and modification (*Fukami, 2015*), drive inter-individual variability. Eco-evolutionary feedbacks in which colonizers continue to adapt to the niche opportunities that arise during the dynamic assembly process ensure that communities end up to be, despite their unpredictable configurations, composed of highly adapted members that stably occupy niches and display resilience and colonization resistance (*Walter, 2015*). Since some level of stochasticity during microbiota assembly would be expected to occur even among monozygotic twins, this framework provides an explanation for their surprisingly high inter-individual microbiota

variation (*Goodrich et al., 2016*; *Goodrich et al., 2014*). In addition, low dispersal decreases rates of immigration, which is predicted to enhance priority effects as average time spans between early and later colonizers are increased (*Fukami, 2015*), consequently increasing β-diversity (*Hubbell, 2001*; *Chase and Myers, 2011*; *Rosindell et al., 2012*). These concepts may provide one explanation for the higher microbiota individuality in people living in industrialized societies (which likely have lower levels of dispersal due to sanitation) when compared to non-industrialized populations (*Martínez et al., 2015*). However, although these concepts are in agreement with much of the observational data on microbiomes, the findings in our study derive from experiments testing only two communities and four specific members, and it will require additional mechanistic, observational, and theoretical studies in a variety of contexts to validate the importance of historical perspective of gut microbiota assembly. Specifically, future research should be targeted at determining to what extent general concepts of ecological theory can be applied to host-associated microbial communities.

Apart from contributing to our basic understanding of gut microbial ecology, this study provides a critical foundation for future experiments that test the impact of colonization order in disease predisposition and whether assembly history can be systematically influenced. Given the importance of historical contingency for gut microbiota assembly, clinical and medical interventions early in life (e. g. antibiotics, C-sections, formula feeding) are likely to have longer lasting consequences, driving not only inter-individual differences but potentially also aberrant patterns of colonization that could potentially be prevented by an adjustment of clinical practices to avoid priority effects (*Sprockett et al., 2018*; *Dominguez-Bello et al., 2016*). Our findings are also relevant for the development of strategies to modulate microbiomes. As the understanding of the health-promoting attributes of gut bacteria continues (*Olle, 2013*), it will be important to evaluate how they can be established more permanently. Once assembled, the gut microbiota is extremely resilient to therapeutic modulations, dietary changes and moderate doses of antibiotics (*Dethlefsen and Relman, 2011*; *Martínez et al., 2013*; *Robinson et al., 2010*), and colonization resistance constitutes a major barrier to introducing beneficial microbes (*Walter et al., 2018*). Our results confirmed recent findings in humans that bacterial strains can be stably established in a climax community if closely related species are absent (*Maldonado-Gómez et al., 2016*). However, if competitors are present, stable persistence cannot be achieved. More permanent persistence can be achieved if microbes are introduced early in life. In addition, the findings demonstrated that early introduction of just a few species can divert the entire trajectory of the microbiota. If such shifts can be introduced reproducibly, then early colonizers could be selected to deliberately control microbiome assembly to obtain predictable outcomes. Although there will likely always be strong stochastic elements in microbiome acquisition, priority effects will favor bacteria that are introduced first, thereby providing an opportunity to potentially prevent aberrant microbiomes, and by doing so, dysbiosis-related diseases.

## Materials and methods

### Mouse husbandry

Germ-free C57BL/6 wild-type (WT) and *Rag1*$^{-/-}$ (which lack mature B and T lymphocytes) mice were born and reared in flexible film isolators and maintained under gnotobiotic conditions at the University of Nebraska-Lincoln. Breeding mice and pups (up to 21 days of age) were fed autoclaved Lab-Diet 5021 (Purina Foods, St. Louis, MO) *ad libitum*. After weaning (at 21 days of age), mice were fed autoclaved LabDiet 5K67 (Purina Foods) *ad libitum*. Germ-free status of the breeding colony was performed routinely as described using culture, microscopy, and PCR (*Bindels et al., 2017*). The Institutional Animal Care and Use Committee of the University of Nebraska-Lincoln approved all procedures involving animals (Project ID 731 and 817).

### Donor mice

To avoid artificial niche opportunities, donor ceca communities from adult wild mice were selected. The ceca selected corresponded to i) a mouse originating from a population of mice derived from a wild mouse population sampled in the Massif Central region of France in 2009 and was subsequently housed at the breeding facility of the Max Plank Institute for Evolutionary Biology for three

generations (*Wang et al., 2014*), referred to in the present manuscript as donor community A, and ii) from a wild mouse caught in Scotland (*Weldon et al., 2015*), which is referred to as donor community B.

## Preparation and standardization of donor inocula

To prepare inocula, frozen ceca were introduced into an anaerobic chamber (Bactron600 Shel Lab, Sheldon Manufacturing INC., Cornelius, OR) to avoid oxygen exposure, and slurries were made by suspending cecal contents in pre-reduced, sterile PBS with 10% glycerol (pH 7) at a dilution of 1:10 and stored at −80°C in aliquots of 500 µl to avoid freeze-thawing. To generate standardized donor cecal inocula, the cecal communities of the wild mice was transferred into germ-free C57BL/6J mice reared at the UNL's Animal Research Facility. This standardization was performed to i) obtain larger volumes of inocula needed for the experiments, ii) determine whether a wild mouse microbiota could be successfully established in C57BL/6J mice under laboratory conditions, and iii) standardize the inocula to the same environmental conditions (food, oxygen exposure at gavage, environmental changes due to laboratory conditions, etc.). To establish microbiomes in mice, 100 µl of either donor A or B were orally gavaged into 8 week old C57BL/6J mice (N = 8 for donor A, and N = 5 for donor B) housed in separate flexible plastic isolators. Four weeks after gavage, mice were euthanized and their ceca harvested. Ceca were excised inside a laminar flow hood with aseptic techniques, and immediately snap-frozen in liquid nitrogen. Cecal slurries of the two donor microbiomes were obtained after pooling ceca from recipient mice under anaerobic conditions and homogenizing contents with sterile, pre-reduced PBS with 10% glycerol (1:10 ratio) as described above. Donor microbiomes were stored in 500 µl aliquots and kept frozen at −80°C until further use. DNA extraction was done according to Martínez and co-workers (*Martínez et al., 2010*), and sequencing of the V5-V6 region of the 16S rRNA (*Krumbeck, 2015*) was done at the University of Minnesota Sequencing Center. Analysis of the 16S rRNA data was done as explained below.

## Sequential inoculation of two complex cecal communities into germ-free mice

The first set of experiments was designed to systematically modify colonization order of two donor cecal microbiomes (A and B; see supplementary file for additional information on how donor inocula were selected and prepared), through three experimental treatment groups. In two of the treatments, donor communities were sequentially inoculated in alternating order (*Figure 1A*), where community A was inoculated first and B second (group A/B) and where community B was inoculated first and A second (group B/A). In a third treatment, both donor communities were inoculated at each time point (AB/AB). Either germ-free C57BL/6 wild-type (WT) or *Rag1*$^{-/-}$ mice were inoculated at 10 ± 2 days after birth by oral gavage (first inoculation), and at 36 ± 2 days after birth (second inoculation). Inoculation by oral gavage consisted of introducing 50 µl of cecal contents into the mouth of each pup using a small-sized syringe. The leftover inoculum was deposited on the parents' fur (for 10-day-old mice), on the mouse's fur (for 36-day-old mice), and on the cage bedding. For the AB/AB group, separate vials containing donor A and B aliquots were introduced into the isolators and mixed right before administering to the mice. Pups were weaned at 21 ± 1 days after birth, and the parents were euthanized to collect their ceca for microbiome analysis; pups were placed into separate cages according to sex. On day 78 after birth (±1 day), mice were euthanized and their ceca harvested and stored at −80°C until DNA extraction. Mice were maintained in isolators for the entire duration of the experiment, with separate isolators for different treatments and genetic backgrounds (WT or *Rag1*$^{-/-}$). Each experiment was done in duplicate (N ≥ 5 per replicate with mice from one or two litters).

## Modification of colonization order of specific bacterial colonizers in ex-germ-free mice

A second set of experiments was performed to specifically modify colonization timing of a cocktail of four bacterial strains during gut microbiota assembly relative to that of a complex microbiota. Four strains autochthonous to the mouse gut microbiota were selected: *Lactobacillus reuteri* Lpuph1 (*Oh et al., 2010*), *Lactobacillus johnsonii* DPPM (*Perez-Muñoz et al., 2014*), *Bacteroides vulgatus* RJ2H1 (a strain isolated from laboratory mice at the University of Nebraska-Lincoln; NCBI accession

number PRJNA78795), and *Clostridium cocleatum* ATCC 29902[T] (*Kaneuchi et al., 1979*). Growth conditions of the individual strains in the cocktail mix is detailed in the section below. The donor cecal inocula used in these experiments were prepared from ceca harvested from 8-week-old conventionally raised C3H/HeN mice (5 mice of the same litter) housed at the University of Nebraska-Lincoln, as described above. Germ-free C57BL/6 wild-type (WT) and *Rag1*[-/-] mice were maintained inside independent, flexible-film isolators until inoculation with C3H/HeN ceca donor microbiota at 10 ± 1 days after birth. At that time, litters of pups and their parents were transferred to individually ventilated cages (IVC) mounted on racks with positive airflow until the end of the experiments. Inoculation of all mice with specific colonizer strains or cecal inocula was done by orally gavaging the pups 50 µl of cell suspension (see previous section), except for 5-day-old mice for which a drop of the inoculum was gently placed on pups' snouts which they leaked, thus avoiding over-handling of the pups and potentially hurting them. This technique proved successful for inoculation of the four specific strains in the mix, as seen in the consistent detection of all four strains in fecal samples. Group I received the mix of specific colonizers at day 5 (while in the isolator and before being moved to the IVC rack and receiving the donor microbiota); Group II at day 15 (5 days after receiving the donor microbiota); and Group III at day 36 (*Figure 5A*). Approximately $10^8$ cells per mouse were administrated for each strain, with the exception of *C. cocleatum* (around $10^5$ cells/ mouse). Fecal samples were collected from each mouse right before the first inoculation (same day), and weekly throughout a 5-week period post-inoculation. For Group I experiments, the first fecal sample was collected from the parents instead of the pups due to their young age (5 days); the remaining five fecal collections were obtained from each pup. For Groups II and III, all fecal pellets were collected directly from the pups/adult animals. Mice were weaned at day 21, at which time parents were removed from the cage, euthanized, and their ceca harvested as previously described. Mice were euthanized at day 78 (±1 day), and their ceca collected for DNA extraction, quantification of the specific strains, and microbiota analysis. Each experiment was done in duplicate (N > 5 in each replicate, from one or two litters).

## Growth conditions of strains used as specific colonizers in cocktail

All bacterial strains were grown under anaerobic conditions at 37°C. *L. reuteri* and *L. johnsoniii* were propagated in MRS media (BD Difco Microbiology, Houston, TX) supplemented with 10 g/L of maltose and 5 g/L of fructose (mMRS), while *C. cocleatum* ATCC 29902 was grown in Reinforced Clostridium media (Oxoid Limited), and *B. vulgatus* on was grown in Tryptone Yeast Glucose broth (TYG) for liquid growth (*Perez-Muñoz et al., 2014*) and on Brain Heart Infusion Agar with 10% Sheep Blood for plating. To generate the inoculation stocks, strains were grown separately, harvested at late exponential phase, washed with reduced PBS under anaerobic conditions, cells were pelleted through centrifugation (3220 $x$ $g$ for 10 min), and then re-suspended in pre-reduced PBS with 10% glycerol (pH 7). Quantitative culture revealed approximately $10^{10}$ CFU/ml, with the exception of *C. cocleatum*, which only contained $10^7$ CFU/ml). Stocks were aliquoted for single-use in the experiments (300 µl), and stored at −80°C until use in mouse experiments.

## Quantification of bacterial strains using qRT-PCR

Quantification of the specific colonizers was done by strain-specific qPCR. Strain-specific primers were designed (*Krumbeck, 2015*) to target genes identified to be unique to *L. reuteri* Lpuph1, *L. johnsonii* DPPM, *C. cocleatum* ATCC 29902[T], and *B. vulgatus* R2JH1 (JGI genome ID numbers: 2506381017, 2606217813, 2574179769, and 2510065017, respectively) by comparing their genomes against a selection of closely related strains. The genes selected as targets encode for nicotinamide mononucleotide transporter in *L. reuteri* Lpuph1 (NCBI Accession ID GCF_000179455.1), a subtilase family protein in *C. cocleatum* ATCC 29902[T] (NCBI GCF_900102365.1), the ORF6C protein domain in *L. johnsonii* DPPM, and the TonB-linked outer membrane protein (SusC/RagA family) [locus tag RJ2H1_00017340] for *B. vulgatus* R2JH1 (according to Joint Genome Institute annotations). Primer sequences and PCR fragment lengths are presented in *Supplementary file 6*. Primers' strain specificity was tested in silico by conducting a BLAST search against the NCBI database. Primers were also tested experimentally against DNA isolated from the C3H/HeN donor microbiota by qPCR, and confirmed to produce no background amplification with the donor community. Cell numbers of strains were quantified by absolute quantification using a standard curve prepared with DNA

isolated from cultures for which cell numbers were determined by quantitative culture. Standard curves were prepared by using pure cultures of each microorganism harvested at late exponential phase, determined by growth curves generated for each strain under anaerobic conditions (*Maldonado-Gómez et al., 2016*; *Krumbeck, 2015*). qPCR was performed using a Bio-Rad C1000 Thermal Cycler instrument (Bio-Rad laboratories, CA), with PCR reaction volumes of 25 µl using Brilliant III Ultra-Fast SYBR Green qPCR master mix (Agilent Technologies, Cedar Creek, TX), 0.8 µM primer concentrations, and 1 µl DNA template. Annealing temperatures of 60°C were used.

## DNA extraction from fecal and gut samples

Fecal and cecal samples were diluted (1:10) in sterile PBS (pH 7). 1 ml and 200 µl aliquots of cecal and fecal dilutions, respectively, were used for DNA extractions. DNA used for qPCR was extracted following a standard phenol-chloroform extraction method (*Martínez et al., 2009*), while DNA used for 16S rRNA gene tag sequencing was extracted with QIAamp DNA Stool Mini kit (QIAGEN, Hilden, Germany) with modifications as described previously (*Martínez et al., 2010*). For both methods, enzymatic and mechanical cell lysis steps were included (*Martínez et al., 2010*; *Martínez et al., 2009*).

## Gut microbiota analysis

Cecal microbiota composition for experiments testing the impact of colonization order using whole cecal microbiota was characterized by sequencing the V5-V6 region of the 16S rRNA (*Krumbeck, 2015*), while the V4 region was sequenced for the experiments testing the impact of time of arrival of specific strains (forward primer: 5'-GTGCCAGCMGCCGCGGTAA-3', reverse primer: 5'-GGACTACHVGGGTWTCTAAT-3'). In both cases, pair-end sequencing was done in the MiSeq Illumina platform (2 × 300; MiSeq Reagent Kit v3). Each dataset was processed independently but following the same protocol. In brief, pair-end reads (2 × 300; MiSeq Reagent Kit v3) were merged and quality controlled with the merge-pairs application of the illumina-utils toolset (*Eren et al., 2013*) with quality check (Q30), removal of primers, removal of sequences with at least one mismatch to the primer sequence, and P parameter = 0.03 (for all other parameters default settings were used). Quality-controlled sequences were subjected to Minimum Entropy Decomposition (MED) analysis (*Eren et al., 2015*), which allows the differentiation of single-nucleotide differences, therefore allowing the highest level of resolution when differentiating members of the two donor communities. Because MED is a non-cluster based algorithm, the term 'bacterial types' is used in the manuscript to refer to the phylogenetic bins assigned by MED, instead of operational taxonomic units that are inherently based on a clustering step. Independent MED analyses were done for the two experimental datasets (given that the ceca microbiota used were different). Taxonomic assignment of the representative sequences of each bacterial type was performed using the RDP Classifier and Seqmatch tools, and verified with BLASTn by the NCBI. Quality-controlled sequences were also binned in a reference-based manner using the RDP MultiClassifier tool (*Wang et al., 2007*) as an independent method of sequence classification.

α and β-diversity distances were calculated after rarefying the number of sequences (20,000 sequences/sample for the experiments introducing the whole ceca experiments in different sequential order, and 12,500 sequences/sample for the experiments testing the importance of time of arrival of specific colonizers). α and β-diversity distances were calculated after rarefying the number of sequences using the QIIME pipeline (version 1.9.1) (*Caporaso et al., 2010*). The datasets generated and analyzed during this study are available under https://figshare.com/s/a1ea177bb39717f13800 and https://figshare.com/s/1cfc381825d63780516e.

## Statistical analysis

Because data obtained from α-diversity measurements were normally distributed, significant differences across treatments with Analysis of Variance (ANOVA) and Tukey's *post-hoc* tests was tested using R (*Core Team, 2014*). Overall community structure differences across treatments were computed with Adonis implemented in R's *vegan* package (*Oksanen, 2013*), which performs a permutational multivariate analysis of variance based on distance matrices. The test was run with 999 permutations using Jaccard and Bray Curtis dissimilarity, and groups in the NMDS plots were displayed using the function ordiellipse. To test the effect of different factors on community structure,

the *envfit* function of the vegan package was used with 999 permutations. Random Forest algorithm was done as previously described (*Maldonado-Gómez et al., 2016*).

For compositional analyses, read counts of a taxon per sample were converted to a $Log_{10}$ scale, after addition of 1 count prior to transformation (avoiding the non-defined $Log_{10}$ of 0 count). To compare differences in taxonomic bin abundances across treatments, linear mixed models with treatment as a fixed effect, and the isolator in which the mice were housed as a random effect, were used in the lme4 package (*Bates et al., 2015*), and post-hoc tests were performed with the *multcomp* package (*Hothorn et al., 2008*) in R. FDR corrections were performed to address multiple testing (false discoveries). FDR adjusted $p < 0.05$ were considered significant. Results are presented as mean ±standard deviation.

## Acknowledgements

We are extremely grateful for the technical expertise and skillful animal husbandry provided by Brandon White and the staff at the University of Nebraska-Lincoln Gnotobiotic Mouse Facility. This project was funded through a National Institute of Health (NIH) grant (5R01GM099525-02). JW acknowledges support through the Campus Alberta Innovates Program (CAIP). AKB and ART acknowledge support from the National Institute of General Medical Sciences of the National Institutes of Health (1P20GM104320) and the University of Nebraska-Lincoln.

## Additional information

### Competing interests

Amanda E Ramer-Tait: Daniel A. Peterson is affiliated with Eli Lilly & Co.. The author has no financial interests to declare. The other authors declare that no competing interests exist.

### Funding

| Funder | Grant reference number | Author |
| --- | --- | --- |
| National Institutes of Health | 5R01GM099525-02 | Andrew Benson<br>Danial Peterson<br>Jens Walter |
| National Institute of General Medical Sciences | 1P20GM104320 | Andrew Benson<br>Amanda E Ramer-Tait |

The funders had no role in study design, data collection and interpretation, or the decision to submit the work for publication.

### Author contributions

Inés Martínez, Conceptualization, Supervision, Funding acquisition, Investigation, Writing—original draft, Project administration, Writing—review and editing; Maria X Maldonado-Gomez, Conceptualization, Data curation, Formal analysis, Supervision, Validation, Investigation, Visualization, Methodology, Writing—original draft, Writing—review and editing; João Carlos Gomes-Neto, Hatem Kittana, Hua Ding, Robert Schmaltz, Roberto Jiménez Cardona, Formal analysis, Investigation, Methodology; Payal Joglekar, Supervision, Investigation, Methodology; Nathan L Marsteller, Investigation, Methodology; Steven W Kembel, Formal analysis, Investigation; Andrew K Benson, Conceptualization, Validation, Writing—review and editing; Daniel A Peterson, Conceptualization, Supervision, Funding acquisition, Methodology, Writing—original draft, Project administration, Writing—review and editing; Amanda E Ramer-Tait, Conceptualization, Supervision, Funding acquisition, Investigation, Project administration; Jens Walter, Supervision, Investigation, Methodology, Project administration, Writing—review and editing

Author ORCIDs
Nathan L Marsteller  https://orcid.org/0000-0001-6243-4312
Amanda E Ramer-Tait  https://orcid.org/0000-0003-0950-7548
Jens Walter  http://orcid.org/0000-0003-1754-172X

Ethics
Animal experimentation: The Institutional Animal Care and Use Committee of the University of Nebraska-Lincoln approved all procedures involving animals (Project ID 731 and 817).

Decision letter and Author response
Decision letter https://doi.org/10.7554/eLife.36521.026
Author response https://doi.org/10.7554/eLife.36521.027

## Additional files

**Supplementary files**
• Supplementary file 1. Abundance (% of total sequences) of bacterial types significantly impacted by colonization order in WT mice. Linear mixed model analysis was done to determine the effect of colonization order on community assembly (treatments A/B, AB/AB, B/A; values for A and B (both donor and parent mice) are given as a reference). Results are presented as mean ±standard deviation.
DOI: https://doi.org/10.7554/eLife.36521.014

• Supplementary file 2. Abundance (% of total sequences) of bacterial types significantly impacted by inoculation time of specific colonizing strains in WT mice, assessed through Random Forest analysis (Random Forest coefficient >2 was considered significant). Results are presented as mean ±standard deviation).
DOI: https://doi.org/10.7554/eLife.36521.015

• Supplementary file 3. Abundance (expressed as percent of total sequences) of bacterial types significantly different between WT and $Rag1^{-/-}$, for experiments when the effect of colonization order of two donor cecal communities on the final community composition was tested. Results are presented as mean ±standard deviation.
DOI: https://doi.org/10.7554/eLife.36521.016

• Supplementary file 4. Abundance (% of total sequences) of bacterial types significantly impacted by colonization order in $Rag1^{-/-}$ and WT mice. Statistical analyses were done for treatments A/B, AB/AB, B/A (A and B are given as a reference) using a linear mixed model with genetic background and colinization order as the main effects. Results are presented as mean ±standard deviation.
DOI: https://doi.org/10.7554/eLife.36521.017

• Supplementary file 5. Abundance (% of total sequences) of bacterial types significantly impacted by inoculation time of specific colonizing strains in $Rag1^{-/-}$ mice, assessed through Random Forest analysis (Random Forest coefficient ≥2 was considered significant). Results are presented as mean ± standard deviation. Taxa in bold were also impacted in WT mice (Supplementary File 2)
DOI: https://doi.org/10.7554/eLife.36521.018

• Supplementary file 6. Primer-pair used for absolute quantification of the specific colonizer strains.
DOI: https://doi.org/10.7554/eLife.36521.019

• Transparent reporting form
DOI: https://doi.org/10.7554/eLife.36521.020

**Data availability**
The datasets generated and analyzed during this study are available under https://figshare.com/s/a1ea177bb39717f13800 and https://figshare.com/s/1cfc381825d63780516e.

The following datasets were generated:

| Author(s) | Year | Dataset title | Dataset URL | Database, license, and accessibility information |
|---|---|---|---|---|
| Inés Martínez | 2018 | Ceca communities inoculated with differential succession dataset | https://figshare.com/s/a1ea177bb39717f13800 | Available at Figshare under a CC0 Public Domain licence (https://figshare.com/). |
| Inés Martínez | 2018 | 4-strain cocktail inoculated at different time points dataset | https://figshare.com/s/1cfc381825d63780516e | Available at Figshare under a CC0 Public Domain licence (https://figshare.com/). |

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
