## [Decision Letter]

Thank you for submitting your article "Experimental evaluation of the importance of colonization history in early-life gut microbiota assembly" for consideration by *eLife*. Your article has been reviewed by Wendy Garrett as the Senior Editor, a Reviewing Editor, and three reviewers. The reviewers have opted to remain anonymous.

The reviewers have discussed the reviews with one another and the Reviewing Editor has drafted this decision to help you prepare a revised submission.

The paper by Martinez et al. aims to study an important and fundamental question of community establishment, and specifically, how order of exposure/colonization effects the final composition of communities. It is tested here in the gut of mice, WT or *Rag1^-/-^*, to start and understand the adaptive immune system's role in community establishment. Together, these studies suggest that the timing of colonization can play a significant role in determining emergent community composition. The reviewers agree that this work is of great interest to the larger microbiome community, with strong implications on the establishment of the human microbial communities as well. The data are extremely interesting, but some additional analyses can lift the manuscript even higher.

Essential revisions:

The sample collection is not 100% clear. Multiple figures refer to "Parents A" and "Parents B", and it's unclear where these samples came from. If it is indeed from the fur of the parents, then we should have had "Parents AB" as well, so it's unclear.

Subsection “Establishing a mouse model using distinct, complex microbial communities to study the importance of colonization history”: The authors state a desire "to avoid microbiomes from animal care facilities", but then select an individual (donor A) from a colony maintained for three generations in such a facility (and, presumably, indoors, consuming standardized chow, etc). Why? Also, in subsection “Donor mice”, it sounds like one mouse gave rise to three generations. Perhaps edit (unless this was a cloning lab). The donor A mouse is described as "wild" but was born and lived in a lab.

The experiment design looks good, but there are key analyses that I would expect to see in the paper. Some appear later but should be presented much earlier. Please include a better analysis of bacterial species found in donor vs. recipients. Analysis of the transmitted bacterial species, not in terms of overall statistics (like Shannon diversity), or PCoA plots, but big heat map(s) that show(s) which strains were transferred and which were not. And/or rank abundance curves with relevant taxa highlighted. The PCoA plots often show different axes (not always named axis 1 and 2), and it feels like there are more interesting results in the full data.

Figure 1B: It seems like data from Parents A/B are missing from both WT and *Rag1^-/-^* experiments. I wonder if these parents had higher alpha-diversity, too (like the pups), because they were exposed to a mixture. Please add if available. (To be clear, I mean the parents of the AB/AB treatment mice, given that the parents of the A/B and B/A treatment mice are shown.)

Figure 1C: How come there is variation in samples A,B in panel C but not in B? How many samples are in each group?

Figure 2 needs a lot of polishing. Axes are not consistent, sizes are not consistent. Clearly, it is not ready for publishing. PCoA axis should have% variance explained added to them.

Subsection “Assessing the importance of colonization history and dispersal limitation towards shaping gut microbiota structure” and Figure 2A: It appears that all of the recipient animals cluster closer to donor A than they do to donor B. Is this consistent with the statement that "B/A mice clustered closer to Donor B"? Is there a significant difference in the clustering of A/B vs. B/A mice with Donor A vs. Donor B? In Figure 2A, parent colors are not distinguishable from A/B, B/A; the x-axis label is missing; Parents A/B are missing.

Figure 2C: I am puzzled by the differences across the various colonization orders. It seems that for A/B and B/A within cage and across cages are not that different, whereas for AB/AB there is a strong difference there. How can this be explained? I am also surprised that there is a significant difference in AB/AB comparing across cages to across isolators (*** P < 0.001). The figure doesn't seem to show that strong difference.

Subsection “Assessing the importance of colonization history and dispersal limitation towards shaping gut microbiota structure”: The Adonis analysis and the distances shown in Figure 2C do not seem to agree. From Figure 2C, it appears the isolator would have the strongest effect, and not the cage. Can you provide more details on your Adonis analysis?

Subsection “Assessing the importance of colonization history and dispersal limitation towards shaping gut microbiota structure”: Nice result. To support that local extinction (within isolators) was by drift and not by selection, could you examine the relative abundance of those types (that disappeared) in the donor communities (i.e., the metacommunity)? Assuming low abundance types are more likely to drift to local extinction.

Subsection “Bacterial types affected by colonization order and mechanisms of priority effects”: Before digging deeper into the order, there are clear species that are absent in one donor, and present in the recipients, and it would be very interesting to understand what these are.

Subsection “Bacterial types affected by colonization order and mechanisms of priority effects”: In the linear mixed model analysis, were cage/isolators variable taken into account? The authors made it clear in the previous section that these are relevant variables, yet they are not mentioned here more.

Subsection “Bacterial types affected by colonization order and mechanisms of priority effects”: I would name these (inhibitory/facilitative effects) the opposite. If a strain is overrepresented when inoculated first, that would not be inhibitory. I find these confusing, maybe you should consider explaining it in more detail?

Subsection “Bacterial types affected by colonization order and mechanisms of priority effects”: Please indicate the total number of cases; the three cherry picked examples all have quite low abundances – less than 1%, less than 0.1% –which raises a concern about detection limits and whether these are just sampling depth effects. Were any highly abundant types affected?

The figures are not sharp enough. Some figures do not convey the message clearly and concisely (like the experimental design figures). E.g. Figure 3A – what are the values shown – abundance or log values? A label is missing. The entire figure needs much polishing as well. All the sizes are different, and it is not well positioned of all panels. PCoA plots should not show clustering of samples (for example, with a circle surrounding each group of samples), as this visualization is misleading to show more separation than what the data support. Overall, figures are missing legend headers, axis names, units explanation, etc.

Subsection “Establishing a mouse model using distinct, complex microbial communities to study the importance of colonization history” and in particular "[…] colonization order is the only experimental variable in the model.": Is host age (and associated factors like weaning status and parental proximity) not also a variable? There is no control in which day 10 A or B is followed by day 36 sham; or day 10 sham is followed by day 36 A or B. Isn't it possible that A and B differ in the degree to which their species can persist at a particular developmental age without the competing community? (Overall, pieces of this paragraph might fit better in the Introduction or Discussion section, rather than Results section.) Likewise, in subsection “Timing of arrival impacts persistence of individual colonizers” (second paragraph), by "timing", do the authors mean host age, or relative to the timing of C3H introduction, or both? Again, controls in which nothing was introduced on day 10 might shed light on this.

Subsection “The adaptive immune system does not contribute to historical contingency in gut microbiota assembly”: Nice result. Why not show the data?

Please standardize the terminology:

– Subsection “Establishing a mouse model using distinct, complex microbial communities to study the importance of colonization history”: "OTUs" are often called "types", sometimes called "nodes" or "OTUs", and said to be potentially differentiated by "only one nucleotide". Please clarify what the entities are, and when they differ from one analysis to another (if they do). Please establish terms clearly and use consistently.

– Figure 1A "day 78", subsection “Gut microbiota of recipient mice shows higher diversity when compared to donors” "week 12" and later "day 72": If these all mean the same thing, then please standardize.

Subsection “Timing of arrival impacts persistence of individual colonizers”: The authors mentioned before that gavaging earlier than 10 days may be harmful, and yet here they are gavaging at age 5 days?

Subsection “Bacterial types affected by colonization order and mechanisms of priority effects” and subsection “Timing of arrival impacts persistence of individual colonizers”: Instead of not showing the data, add a supplementary figure.

Supplementary file 1: What do the "a", "b", "ab" symbols mean?

Subsection “Assessing the importance of colonization history and dispersal limitation towards shaping gut microbiota structure”, regarding "niche-related differences should be marginal in our study": This is confusing, because it would seem that unweaned and weaned mouse gut environments, for example, must be quite different in terms of niches. There's no control for developmental age, and we don't know how A or B act individually based on timing, it seems? (This repeats an above comment.) What might we expect to see had the experiments been performed in adult germ-free animals?

Subsection “Bacterial types affected by colonization order and mechanisms of priority effects”: The statement that "bacterial functions among gut microbiota members are not necessarily related to phylogeny" is quite bold based on the evidence provided. The authors should consider defining "functions" more clearly or providing evidence that horizontal gene transfer (rather than phylogeny) does explain bacterial functions if they believe this to be true.

Subsection “Bacterial types affected by colonization order and mechanisms of priority effects” and Figure 3C: These results are fascinating, please discuss more. It seems like in some cases (left panels) it's really a mutual exclusive abundance of the two types, whereas in the right panels, it seems like these could co-exist, like in the AB/AB samples.

Subsection “Timing of arrival impacts persistence of individual colonizers”: The logical path/flow from experiment 1 (donor A, B) to experiment 2 (4-strain, C3H) is unclear. There's a lack of integration between the two; like two separate studies with similar themes placed side by side. What findings from the first prompted the second? What rationale?

Subsection “Timing of arrival impacts persistence of individual colonizers”: This is a really interesting result. Does it have anything to do with the microbial composition of the pooled cecal samples used for the day 10 gavage. Do you have any data on these pooled communities? Do they have any of these species tested here?

Subsection “Timing of arrival impacts persistence of individual colonizers”: It seems that there are several non-exclusive alternate explanations that are also consistent with the observations, given the very small number of species tested. For example, one could imagine that the timing of species introduction relative to innate immune system development could be a contributing factor, or that non-identical species also contribute to competitive exclusion.

Subsection “The adaptive immune system does not contribute to historical contingency in gut microbiota assembly”: The word "adapt" is used loosely here. Adaptation through monopolization – what does that mean? The host (immune system) can respond and adapt during postnatal development.

Discussion section: It is striking that 19/20 species that are effected by colonization order in WT mice show the same behavior in *Rag1^-/-^* mice, suggesting that this activity is coded into their genomes – does this make it a deterministic process? Can the authors provide any insight into the features in the genomes of these ~20 species that likely contribute to this result?

Although the discussion is already quite long, perhaps some sections could be condensed in order to clarify the limitations of the sample sizes used (2 communities, 4 species) in deriving general principles. This would be helpful for a general audience.

The paper seems to be inconsistent in whether the effects observed are large or small. E.g., "Given the importance of historical contingency for gut microbiota assembly, clinical and medical interventions early in life (e.g., antibiotics, C-sections, formula feeding) are likely to have longer lasting consequences." vs. "Although the relative importance of assembly history appears small in our experiments (and less than that of dispersal limitation)[…]" For general readers, it would be helpful to be more quantitative about the magnitude of these changes.

Other examples from subsection “The adaptive immune system does not contribute to historical contingency in gut microbiota assembly”:

– Quote "significantly lower": Looking at the plot, the effect seems quite subtle.

– Larger effect sizes/more bacterial types in what compared to what? This is confusing.

– In the last paragraph, "both were increased" in *Rag1^-/-^*: I’m not sure given the data, and I’m not sure you agree, given your Abstract.

Many typos include:

"stains": strains

"locus": loci

"-wide wide": -wide

"importance": importance of

"ceca": cecum

"this a finding": this finding

"time a": time of a

"bacterial types bacterial types": bacterial types

"types to significantly differ": types that significantly differed

"inoculums": inocula

"impact colonization": impact of colonization

"assignation": assignment

"B-specific of the donor": B-specific types of the donor

---

## [Author Response]

The paper by Martinez et al. aims to study an important and fundamental question of community establishment, and specifically, how order of exposure/colonization effects the final composition of communities. It is tested here in the gut of mice, WT or Rag1^-/-^, to start and understand the adaptive immune system's role in community establishment. Together, these studies suggest that the timing of colonization can play a significant role in determining emergent community composition. The reviewers agree that this work is of great interest to the larger microbiome community, with strong implications on the establishment of the human microbial communities as well. The data are extremely interesting, but some additional analyses can lift the manuscript even higher.

We are grateful for this overall positive and encouraging assessment, and we thank the reviewers for extremely constructive reviews.

Essential revisions:The sample collection is not 100% clear. Multiple figures refer to "Parents A" and "Parents B", and it's unclear where these samples came from. If it is indeed from the fur of the parents, then we should have had "Parents AB" as well, so it's unclear.

We agree that this was confusing and thank the reviewer for pointing it out. We have now modified Figure 1A (experimental design), which is now a separate figure (Figure 1), to clarify how and when samples were collected and how samples are called, including the samples from the parents. In addition, the text in the manuscript was modified to specify when the parent samples were collected (subsection “Establishing a mouse model using distinct, complex microbial communities to study the importance of colonization history”).

Subsection “Establishing a mouse model using distinct, complex microbial communities to study the importance of colonization history”: The authors state a desire "to avoid microbiomes from animal care facilities", but then select an individual (donor A) from a colony maintained for three generations in such a facility (and, presumably, indoors, consuming standardized chow, etc). Why? Also, in subsection “Donor mice”, it sounds like one mouse gave rise to three generations. Perhaps edit (unless this was a cloning lab). The donor A mouse is described as "wild" but was born and lived in a lab.

In the revised manuscript, we have now clarified that what we wanted was to avoid mice from commercial facilities that are often aberrant (subsection “Establishing a mouse model using distinct, complex microbial communities to study the importance of colonization history”). In addition, we now strictly refer to the mice that were caught in the wild and then housed in a facility as “wild-derived” (subsection “Establishing a mouse model using distinct, complex microbial communities to study the importance of colonization history”), and we have clarified how these mice were derived (subsection “Preparation and standardization of donor inocula”).

The experiment design looks good, but there are key analyses that I would expect to see in the paper. Some appear later but should be presented much earlier. Please include a better analysis of bacterial species found in donor vs. recipients. Analysis of the transmitted bacterial species, not in terms of overall statistics (like Shannon diversity), or PCoA plots, but big heat map(s) that show(s) which strains were transferred and which were not. And/or rank abundance curves with relevant taxa highlighted. The PCoA plots often show different axes (not always named axis 1 and 2), and it feels like there are more interesting results in the full data.

We thank the reviewer for this suggestion. In fact, a heat map was provided (previously Figure 3A) that does in fact show the abundances of all bacterial types in both donor and recipients, but a clear legend was missing, and we apologize for that. This has now been corrected. The heat map provides clear information on the bacterial types found in donor vs. Recipient. We agree with the reviewer that the heat map should be presented earlier, and we now show the heat map in Figure 2A and have provided text earlier on (subsection “Gut microbiota of recipient mice shows higher diversity when compared to donors”) to present to what degree types were transferred. We further discuss how the relative abundances shown in the heatmap is similar in donor and recipient mice for most types, demonstrating that the abundance distribution of bacterial types in the recipient mice reflected to a large degree those of the donors, showing that transfer of the microbiota did not result in major rearrangements in community structure.

We prefer not to focus on the exact taxonomic assignments of relevant taxa as such assignments are difficult to make for these largely uncharacterized wild-derived microbiomes. Please be aware that we do make such assignments for the taxa that are impacted by colonization order (Supplementary file 1), which is the focus of this study. Clearly, both rank abundance curves and larger tables could be provided if requested, but we do question that value of adding this information.

Figure 1B: It seems like data from Parents A/B are missing from both WT and Rag1^-/-^ experiments. I wonder if these parents had higher alpha-diversity, too (like the pups), because they were exposed to a mixture. Please add if available. (To be clear, I mean the parents of the AB/AB treatment mice, given that the parents of the A/B and B/A treatment mice are shown.)

Indeed, the Parents A/B also have higher alpha-diversity consistent with the findings in all recipient mice, and we have added them into the revised Figures 2B-E. Please be aware that parents were not included in the statistical analysis as they only served as a reference and the objective of the statistical test was to determine if recipient mice at 78 days were more diverse than donor mice.

Figure 1C: How come there is variation in samples A,B in panel C but not in B? How many samples are in each group?

The reason for this is that to determine A and B specific types, data from donor and parent mice were combined to avoid sample size bias that results from having only one sample profiled for each of the donor communities. Amount of samples are now added to the figure legend (now Figure 2), and the figure legend has been modified to clarify these issues.

Figure 2 needs a lot of polishing. Axes are not consistent, sizes are not consistent. Clearly, it is not ready for publishing. PCoA axis should have% variance explained added to them.

Figure 2, which is now Figure 3 in the revised version, has been polished to make axes and sizes consistent. Please be aware that these are not PCoA plots, but NMDS plots (non-metric multidimensional scaling, within the vegan package in R), which have many advantages of PCoA, such as making assumption about data distribution, and is a very popular method in ecology. There is no meaningful way to express axis variation in NMDS. Instead, a low stress value is preferred (as a guide <0.05 stress value is considered excellent, and <0.1 good). Stress values have been added to the figures.

Subsection “Assessing the importance of colonization history and dispersal limitation towards shaping gut microbiota structure” and Figure 2A: It appears that all of the recipient animals cluster closer to donor A than they do to donor B. Is this consistent with the statement that "B/A mice clustered closer to Donor B"? Is there a significant difference in the clustering of A/B vs. B/A mice with Donor A vs. Donor B? In Figure 2A, parent colors are not distinguishable from A/B, B/A; the x-axis label is missing; Parents A/B are missing.

We thank the reviewer for pointing this out and agree it was not clear. We have now clarified that we mean distances between the A/B and the B/A group in comparison with the other groups, and not absolute distances (subsection “Assessing the importance of colonization history and dispersal limitation towards shaping gut microbiota structure”). Since this is only an overview graph that also contains donor and parent mice, statistics were not done here. However, they are done in the analysis shown in Figure 3C without the parents, and we have now compared average distances of the groups to the donors A and B, and they indeed are significantly different (Figure 3B). Colors have changed to make them distinguishable.

Figure 2C: I am puzzled by the differences across the various colonization orders. It seems that for A/B and B/A within cage and across cages are not that different, whereas for AB/AB there is a strong difference there. How can this be explained?

This is an interesting observation, but our experiment does not allow conclusions on why that is. The important massage is that, although all of these mice received the same communities, they clearly cluster according to barriers of dispersal, which is discussed in the text.

I am also surprised that there is a significant difference in AB/AB comparing across cages to across isolators (*** P < 0.001). The figure doesn't seem to show that strong difference.

We thank the reviewer for pointing this out. The difference is significant, but indeed, the p value was only 0.006, which makes it ** P < 0.01. This mistake has been corrected in what is now Figure 3D.

Subsection “Assessing the importance of colonization history and dispersal limitation towards shaping gut microbiota structure”: The Adonis analysis and the distances shown in Figure 2C do not seem to agree. From Figure 2C, it appears the isolator would have the strongest effect, and not the cage. Can you provide more details on your Adonis analysis?

The Adonis test is evaluating what is the variable that drives similarities across the samples, and the better explanatory variable is the factor ‘cage’, as the microbiomes within cages are more similar to each other than across isolators (which is consistent with what Figure 3D is showing). The text was changed from “explaining most of the variation” to “variables revealed that the data was better explained by the factor cage (P < 0.001, R^2^ = 0.66), followed by isolator (P < 0.001, R^2^ = 0.46) and colonization order (P < 0.001, R^2^ = 0.21).” (subsection “Assessing the importance of colonization history and dispersal limitation towards shaping gut microbiota structure”).

Subsection “Assessing the importance of colonization history and dispersal limitation towards shaping gut microbiota structure”: Nice result. To support that local extinction (within isolators) was by drift and not by selection, could you examine the relative abundance of those types (that disappeared) in the donor communities (i.e., the metacommunity)? Assuming low abundance types are more likely to drift to local extinction.

Clearly, such an analysis could be done. However, given that dispersal limitation is the only factor that contributes to the clustering, we are pretty confident that we are in fact dealing with drift here. Second, the focus of our work is to identify the importance of historical processes, and we therefore think that such an analysis would be out of scope and confuse more than help. Clearly, if requested, it could be added.

Subsection “Bacterial types affected by colonization order and mechanisms of priority effects”: Before digging deeper into the order, there are clear species that are absent in one donor, and present in the recipients, and it would be very interesting to understand what these are.

Supplementary file 1 does show all 20 types affected by colonization order, many of which cannot be classified to species level. Also, as described above, only 7% of the types detected in the recipient mice are not detectable in the donors, and they only make up 0.7% of the total sequences. This shows that only a very small proportion of the recipient microbiome cannot be traced back to donors, and as described above, this is now included in the manuscript (subsection “Gut microbiota of recipient mice shows higher diversity when compared to donors”). We do therefore not think it would be helpful to focus space here on the discussion of the types that are absent in the donor and present in the recipients.

Subsection “Bacterial types affected by colonization order and mechanisms of priority effects”: In the linear mixed model analysis, were cage/isolators variable taken into account? The authors made it clear in the previous section that these are relevant variables, yet they are not mentioned here more.

It is mentioned now in the text that 'isolator' was indeed used as a random variable in the model. We think that it is not appropriate to use both ‘isolators’ and ‘cages’ as the two variables are dependent. However, we have also done the analysis using 'cages' as the random variable, and out of the 20 bacterial types detected with 'isolators' as the random variable, 18 were also detected with 'cages' as the random variable. 'Cages' as the random variable led to a higher number of significant bacterial types to be detected as different across groups, but visual inspection of the distributions indicated that the findings are less robust for the extra types detected. As a conservative approach, we decided to present results using 'isolators' as the random variable.

Subsection “Bacterial types affected by colonization order and mechanisms of priority effects”: I would name these (inhibitory/facilitative effects) the opposite. If a strain is overrepresented when inoculated first, that would not be inhibitory. I find these confusing, maybe you should consider explaining it in more detail?

We have included a more detailed explanation in the revised manuscript to argue that overrepresentation when inoculated first does indeed point to inhibition in the subsection “Bacterial types affected by colonization order and mechanisms of priority effects”.

Subsection “Bacterial types affected by colonization order and mechanisms of priority effects”: Please indicate the total number of cases; the three cherry picked examples all have quite low abundances – less than 1%, less than 0.1% – which raises a concern about detection limits and whether these are just sampling depth effects. Were any highly abundant types affected?

Two of the examples that we present have percentages of around 1%. Given that there are around three hundred bacterial types in these communities, we do not consider a 1% relative abundance ‘low abundance’. In addition, we sequenced to a depth of 20,000 sequences per sample. We therefore think it is unlikely that our observations are purely through sampling. Lastly, we show averages obtained in at least 12 mice per group and two independent experiments. We therefore argue that random effects through sampling depth would not have been that consistent and would not have resulted in statistically significant findings.

The figures are not sharp enough. Some figures do not convey the message clearly and concisely (like the experimental design figures). E.g. Figure 3A – what are the values shown – abundance or log values? A label is missing. The entire figure needs much polishing as well. All the sizes are different, and it is not well positioned of all panels. PCoA plots should not show clustering of samples (for example, with a circle surrounding each group of samples), as this visualization is misleading to show more separation than what the data support. Overall, figures are missing legend headers, axis names, units explanation, etc.

The figures were actually quite sharp, so we think this was a problem of the Pdf that was generated for the review purpose. Figure 3A, which is now Figure 2A, has been changed, and legend has been added. Figures were extensively polished. As we write above, these are NMDS plots, and circles are quite standard here, and we have added information to the Material and Methods on how these circles were generated (subsection “Statistical analysis”). Axis names and explanations have been added.

Subsection “Establishing a mouse model using distinct, complex microbial communities to study the importance of colonization history” and in particular "[…] colonization order is the only experimental variable in the model": Is host age (and associated factors like weaning status and parental proximity) not also a variable?

Host age and weaning status were strictly standardized and not different between treatment groups. We acknowledge that there will be developmental changes in the offspring mice both in this paragraph and in the discussion. These are factors that could account for some of our findings, and this is discussed (see Discussion section), but we argue that these are not variables among groups.

There is no control in which day 10 A or B is followed by day 36 sham; or day 10 sham is followed by day 36 A or B. Isn't it possible that A and B differ in the degree to which their species can persist at a particular developmental age without the competing community? (Overall, pieces of this paragraph might fit better in the Introduction or Discussion section, rather than Results section.) Likewise, in subsection “Timing of arrival impacts persistence of individual colonizers” (second paragraph), by "timing", do the authors mean host age, or relative to the timing of C3H introduction, or both? Again, controls in which nothing was introduced on day 10 might shed light on this.

We acknowledge that developmental differences at day 10 might contribute to historical contingency, and this is discussed. However, we find it unlikely that this is due to differences in persistence at early age. Please be aware that in our experiments, the parent mice also get colonized (and this was confirmed by sequencing), which allows vertical and horizontal transmission (e.g. through coprohagy) to the offspring up until weaning and thus maturity, which makes it unlikely that effects are only because species can persist when colonizing early. We have added additional information to the manuscript (subsection “Establishing a mouse model using distinct, complex microbial communities to study the importance of colonization history”) to emphasize this feature of our model. We would further like to add that even if developmental differences contribute to historical contingency, colonization order would still be important microbiome assembly. This is also discussed in the manuscript (Discussion section). What is meant by ‘arrival timing’ (it is relative to competitors arrival) has been clarified (subsection “Timing of arrival impacts persistence of individual colonizers”).

Subsection “The adaptive immune system does not contribute to historical contingency in gut microbiota assembly”: Nice result. Why not show the data?

In Supplementary file 4, we do label the taxa that are shared between WT and *Rag1^-/-^* mice.

Please standardize the terminology:– Subsection “Establishing a mouse model using distinct, complex microbial communities to study the importance of colonization history”: "OTUs" are often called "types", sometimes called "nodes" or "OTUs", and said to be potentially differentiated by "only one nucleotide". Please clarify what the entities are, and when they differ from one analysis to another (if they do). Please establish terms clearly and use consistently.

This has been done.

– Figure 1A "day 78", subsection “Gut microbiota of recipient mice shows higher diversity when compared to donors” "week 12" and later "day 72": If these all mean the same thing, then please standardize.

This has now been standardized.

Subsection “Timing of arrival impacts persistence of individual colonizers”: The authors mentioned before that gavaging earlier than 10 days may be harmful, and yet here they are gavaging at age 5 days?

5 day mice were not gavaged. We have now clarified this in the Material and Methods (subsection “Modification of colonization order of specific bacterial colonizers in ex-germ free mice”).

Subsection “Bacterial types affected by colonization order and mechanisms of priority effects” and subsection “Timing of arrival impacts persistence of individual colonizers”: Instead of not showing the data, add a supplementary figure.

For subsection “Assessing the importance of colonization history and dispersal limitation towards shaping gut microbiota structure”, we would prefer not to show non-significant findings at phyla, family, and genera level, but could do so if requested. For subsection “Timing of arrival impacts persistence of individual colonizers”, we have now included the exact results to the text.

Supplementary file 1: What do the "a", "b", "ab" symbols mean?This has now been included.Subsection “Assessing the importance of colonization history and dispersal limitation towards shaping gut microbiota structure”, regarding "niche-related differences should be marginal in our study": This is confusing, because it would seem that unweaned and weaned mouse gut environments, for example, must be quite different in terms of niches.

We have clarified that we refer to niche-related differences when mice are compared (subsection “Assessing the importance of colonization history and dispersal limitation towards shaping gut microbiota structure”).

There's no control for developmental age, and we don't know how A or B act individually based on timing, it seems? (This repeats an above comment.) What might we expect to see had the experiments been performed in adult germ-free animals?

As we responded above, mice are colonized during a time window of 10 days to weaning (21 days), in which they are exposed to the microbiota of their parents (which are also colonized). We therefore do not find it likely that differences are just due to developmental age. This has been clarified in the revised manuscript (subsection “Establishing a mouse model using distinct, complex microbial communities to study the importance of colonization history”).

Subsection “Bacterial types affected by colonization order and mechanisms of priority effects”: The statement that "bacterial functions among gut microbiota members are not necessarily related to phylogeny" is quite bold based on the evidence provided. The authors should consider defining "functions" more clearly or providing evidence that horizontal gene transfer (rather than phylogeny) does explain bacterial functions if they believe this to be true.

It is pretty well established that through convergent evolution, functions and traits can be similar in taxa that are not necessarily related. This fact is more important in bacteria due to Horizontal Gene Transfer. Not only can taxa that are only distantly related act similar, strains among the same species can be hugely different. Since all of this is well established, we argue that this statement does not need further qualification, but we could provide references if requested. Just as an example, consider *E. coli* Nissle vs. EHEC, which are both the same species; one can act as a probiotic while the other one might kill you.

Subsection “Bacterial types affected by colonization order and mechanisms of priority effects” and Figure 3C: These results are fascinating, please discuss more. It seems like in some cases (left panels) it's really a mutual exclusive abundance of the two types, whereas in the right panels, it seems like these could co-exist, like in the AB/AB samples.

We thank the reviewer for this observation. It is indeed fascinating, and we have added two sentences to the manuscript to discuss this more (subsection “Bacterial types affected by colonization order and mechanisms of priority effects”).

Subsection “Timing of arrival impacts persistence of individual colonizers”: The logical path/flow from experiment 1 (donor A, B) to experiment 2 (4-strain, C3H) is unclear. There's a lack of integration between the two; like two separate studies with similar themes placed side by side. What findings from the first prompted the second? What rationale?

We completely agree, and we have included text to provide clear rationale for the second line of experiments (subsection “Timing of arrival impacts persistence of individual colonizers”).

Subsection “Timing of arrival impacts persistence of individual colonizers”: This is a really interesting result. Does it have anything to do with the microbial composition of the pooled cecal samples used for the day 10 gavage. Do you have any data on these pooled communities? Do they have any of these species tested here?

It might have escaped the reviewer’s attention that this was covered in detail (subsection “Timing of arrival impacts persistence of individual colonizers” in the revised manuscript).

Subsection “Timing of arrival impacts persistence of individual colonizers”: It seems that there are several non-exclusive alternate explanations that are also consistent with the observations, given the very small number of species tested. For example, one could imagine that the timing of species introduction relative to innate immune system development could be a contributing factor, or that non-identical species also contribute to competitive exclusion.

We agree and we have added a qualifier to our conclusion (subsection “Timing of arrival impacts persistence of individual colonizers”).

Subsection “The adaptive immune system does not contribute to historical contingency in gut microbiota assembly”: The word "adapt" is used loosely here. Adaptation through monopolization – what does that mean? The host (immune system) can respond and adapt during postnatal development.

We have now defined monopolization (subsection “Bacterial types affected by colonization order and mechanisms of priority effects”), and it is in fact an adaptive evolutionary process. We still think that the term ‘adapt’ is good when it refers to the response of the immune system, as we specifically test the role of the “adaptive” immune system with our experiments in *Rag1^-/-^* mice. This arm of the immune system does in fact ‘adapt’ to the immunological challenge.

Discussion section: It is striking that 19/20 species that are effected by colonization order in WT mice show the same behavior in Rag1^-/-^ mice, suggesting that this activity is coded into their genomes – does this make it a deterministic process? Can the authors provide any insight into the features in the genomes of these ~20 species that likely contribute to this result?

The ability to benefit from earlier colonization is, as the reviewer says, likely encoded in their genome. Species that are able to preempt niches or monopolize (adapt) will be benefiting from priority effects. However, most bacteria will have these abilities at least to some degree, and if priority effects arise will be highly dependent on the competing communities, e.g. the presence of competitors with niche overlap. We would therefore prefer not to perform such a genomic analysis as we think this would be highly confounded (a member not benefiting from priority effects might well do so in a different context). In addition, many aspects of priority effects are based on selection and competition but, in ecology, the process is still not considered completely deterministic as the order of arrival does influence the interactions between the members, and this order is, and many of the mechanisms (e.g. monopolization, which is based on mutations), in most cases, stochastic.

Although the discussion is already quite long, perhaps some sections could be condensed in order to clarify the limitations of the sample sizes used (2 communities, 4 species) in deriving general principles. This would be helpful for a general audience.

We have included another two sentences on the limitations of our study (Discussion section).

The paper seems to be inconsistent in whether the effects observed are large or small. E.g., "Given the importance of historical contingency for gut microbiota assembly, clinical and medical interventions early in life (e.g., antibiotics, C-sections, formula feeding) are likely to have longer lasting consequences." vs. "Although the relative importance of assembly history appears small in our experiments (and less than that of dispersal limitation)[…]". For general readers, it would be helpful to be more quantitative about the magnitude of these changes.

We do not think we are inconsistent as we are very clear that even the small effects that are detected in our experiments with a small number of species would be amplified under real life conditions in which hundreds of species are likely to be stochastically acquired (Discussion section).

Other examples from subsection “The adaptive immune system does not contribute to historical contingency in gut microbiota assembly”:– Quote "significantly lower": Looking at the plot, the effect seems quite subtle.

We added that *Rag1^-/-^* had a ‘slight’ but significantly lower number of types to indicate that the effects was indeed subtle.

– Larger effect sizes/more bacterial types in what compared to what? This is confusing.

This has been fixed.

– In the last paragraph, "both were increased" in Rag1^-/-^: I’m not sure given the data, and I’m not sure you agree, given your Abstract.

We do think that our data supports the higher importance of colonization history in *Rag1^-/-^*, e.g. through showing that all four specific colonizers were affected and that day 14 colonization altered trajectories of the microbiome when compared to day 36, which both did not happen in WT mice. However, the reviewer is correct that in some instances, the importance of colonization history was reduced. We have changed the Abstract, parts of the Results section and the Discussion section to be as clear as possible with our conclusions.

Many typos include:"stains": strains"locus": loci"-wide wide": -wide"importance": importance of"ceca": cecum"this a finding": this finding"time a": time of a"bacterial types bacterial types": bacterial types"types to significantly differ": types that significantly differed"inoculums": inocula"impact colonization": impact of colonization"assignation": assignment"B-specific of the donor": B-specific types of the donor

This has all been fixed, we have worked extensively with the manuscript to remove additional typos.